# RELATIVE ENTROPY GRADIENT SAMPLER FOR UNNORMALIZED DISTRIBUTIONS

## ABSTRACT

We propose a relative entropy gradient sampler (REGS) for sampling from unnormalized distributions. REGS is a particle method that seeks a sequence of simple nonlinear transforms iteratively pushing the initial samples from a reference distribution into the samples from an unnormalized target distribution. To determine the nonlinear transforms at each iteration, we consider the Wasserstein gradient flow of relative entropy. This gradient flow determines a path of probability distributions that interpolates the reference distribution and the target distribution. It is characterized by an ODE system with velocity fields depending on the density ratios of the density of evolving particles and the unnormalized target density. To sample with REGS, we need to estimate the density ratios and simulate the ODE system with particle evolution. We propose a novel nonparametric approach to estimating the logarithmic density ratio using neural networks. Extensive simulation studies on challenging multimodal 1D and 2D mixture distributions and Bayesian logistic regression on real datasets demonstrate that the REGS outperforms the state-of-the-art sampling methods included in the comparison.

## 1 INTRODUCTION

Sampling from unnormalized distributions plays a fundamental role in statistical inference and machine learning. This problem is frequently encountered in Bayesian statistics. Conducting Bayesian analysis requires evaluation of multi-dimensional integrals where analytical expressions for unnormalized posterior distributions are usually not available. Consequently, sampling is necessary for Monte Carlo approximation of these integrals. In this work, we propose a general purpose sampling algorithm for unnormalized distributions.

Markov chain Monte Carlo (MCMC) methods (Andrieu et al., 2003; Brooks et al., 2011) are widely used to sample from unnormalized distributions. Sampling with MCMC relies on defining an appropriate transition kernel to construct a Markov chain whose equilibrium distribution is precisely the target distribution. Based on rejection sampling, the Metropolis–Hastings algorithm (Metropolis et al., 1953; Hastings, 1970; Tierney, 1994; Dunson & Johndrow, 2019) provides a flexible framework for general MCMC sampling. To implement a Metropolis–Hastings algorithm, one needs to specify a proposal density and an acceptance policy. However, without a careful design of these two aspects, the Metropolis–Hastings algorithm can be inefficient due to strong correlations, slow mixing, or low acceptance rates, especially in the large-scale and high-dimensional settings. Moreover, proposals through discretizing some continuous processes like Langevin diffusion and Hamiltonian dynamics are introduced (Roberts & Tweedie, 1996; Roberts & Stramer, 2002; Duane et al., 1987; Neal, 2011; Hoffman & Gelman, 2014) and further enhanced by stochastic gradient estimation (Welling & Teh, 2011; Chen et al., 2014).

Variational Bayesian inference (Beal, 2003), often simply referred to as variational inference (VI) (Wainwright & Jordan, 2008; Blei et al., 2017), is another prominent approach to sampling from unnormalized distributions. VI approximates the unnormalized posterior distribution with a restricted parametric variational posterior distribution by minimizing the Kullback-Leibler (KL) divergence between them. Since the true posterior distribution is intractable, VI turns to maximize a surrogate variational objective called the evidence lower bound (ELBO). However, one is required to trade off the parameterization flexibility of variational posteriors against the optimization complexity of ELBO in practice.

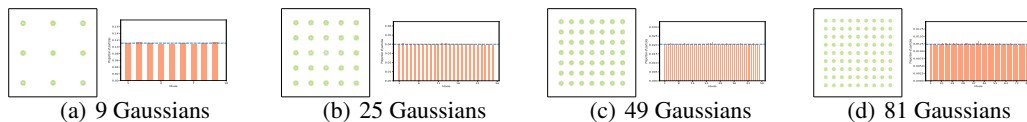

Figure 1: Scatter plots of generated samples and histograms of generated sample counts according to the nearest neighbor mode by REGS for mixtures of **9, 25, 49**, and **81** Gaussians with equal weights. As the plots indicate, generated samples by REGS cover every component of the mixture distributions and are nearly equally allocated to all components.

In the spirit of VI, particle-based variational inference (ParVI) (Liu & Wang, 2016; Chen et al., 2018; Zhu et al., 2020) iteratively optimizes a set of particles to mimic a functional gradient descent for minimizing the KL divergence. ParVI seeks to move a variational distribution towards the unnormalized target distribution, along a steepest descent direction of the KL divergence. In a continuous view, these movements of variational distributions can be understood as a gradient flow in probability measure spaces (Liu et al., 2019a;b). A key part of ParVI is how to estimate the desired steepest descent direction (i.e., functional gradient) from the evolving random particles. An elegant approach is the Stein variational gradient descent (SVGD) Liu & Wang (2016). In SVGD , the functional gradient descent is embedded in a reproducing kernel Hilbert space (RKHS), which is further recognized as a gradient flow under the Stein geometry (Liu, 2017; Lu et al., 2019; Duncan et al., 2019). A drawback of SVGD is that it tends to collapse at part of the modes of the target, due to a negative correlation between the data dimensionality and the repulsive force in the RKHS (Zhuo et al., 2018).

In this work, we propose a relative entropy gradient sampler (REGS) for sampling from unnormalized target distributions. To approximate a target distribution, we consider the Wasserstein gradient flow of relative entropy (or KL divergence), named relative entropy gradient flow. The relative entropy gradient flow represents a path of probability distributions that follows the functional gradient descent direction of relative entropy. There exists an ODE system of random particles that uniquely determines the spatial and temporal dynamics of the relative entropy gradient flow. Therefore, to sample with REGS, we only need to simulate the ODE system with particle evolution. Evaluating the velocity fields of this ODE system can be transformed into estimating the logarithmic density ratio between the density of evolving particles and the unnormalized target density. Based on this observation, we propose a novel logarithmic density ratio estimation method for unnormalized distributions. By alternating between particle evolution and velocity field estimation, we can collect a set of stable particles which are approximately distributed as the target distribution. Our contributions can be summarized as follows:

(1) Building upon the relative entropy gradient flow, we propose the relative entropy gradient sampler (REGS) for unnormalized target distributions. REGS preserves high efficiency and strong stability with respect to increasing singularity in mixtures of Gaussians, when the number of components increases (as shown in Figure 1), the variance of each component decreases, and the distance between any two components increases.

(2) We propose to directly estimate velocity fields of the relative entropy gradient flow as gradients of logarithmic density ratios, that is computationally stable and efficient.

(3) We develop a nonparametric approach to estimating the density ratio between an unnormalized density and an underlying density represented by samples, which is of independent interest.

(4) We present experimental comparisons on varieties of multi-mode synthetic data and benchmark data and demonstrate that REGS is a more accurate sampler than the popular samplers including ULA, MALA and SVGD.

**Related work** The proposed REGS is most related to sampling methods based on the relative entropy gradient flow, in particular, the recently proposed SVGD (Liu & Wang, 2016; Liu, 2017), which estimates the velocity fields of the relative entropy gradient flow in a reproducing kernel Hilbert space. See also Korba et al. (2020); Salim et al. (2021; 2020) for theoretical analysis of SVGD. In contrast, REGS approximates the velocity fields based on a novel logarithmic density ratio estimation approach with deep neural networks. The undesirable mode collapse feature of SVGD is not inevitable for

REGS since the approximation and expressive powers of deep neural networks is known to surpass those of kernel methods.

MCMC algorithms constructed from overdamped Langevin diffusion can be studied as discretization of the relative entropy gradient flow (Jordan et al., 1998). Based on the Euler-Maruyama discretization of overdamped Langevin diffusion, unadjusted Langevin algorithm (ULA) (Roberts & Tweedie, 1996) aims at generating samples from an approximation of the unnormalized target, but is biased for fixed step size. When a Metropolis-Hastings step is included, Metropolis-adjusted Langevin algorithm (MALA) (Roberts & Tweedie, 1996) is capable of correcting the bias, but leaves a large number of intermediate samples rejected. In REGS, one only needs to estimate the deterministic velocity fields of the relative entropy gradient flow, which differs from running ULA and MALA with randomness from diffusion processes. All particles produced by REGS are generated from an approximation of the target distribution.

Another line of work (Gao et al., 2019; 2021) uses Wasserstein gradient flows of $f$-divergences for generative learning with samples from the underlying target distribution. In their work, evaluating velocity fields of gradient flows also boils down to estimating density ratios. However, our current problem is to sample from an unnormalized target density. Furthermore, we propose a novel density ratio estimation procedure when the target distribution is only known up to a normalizing constant.

**Notation** Let $\mathcal{P}_2(\mathcal{X})$ be the space of Borel probability measures on a support space $\mathcal{X} \subset \mathbb{R}^d$ with a finite second moment, and let $\mathcal{P}_2^a(\mathcal{X})$ be a subspace of $\mathcal{P}_2(\mathcal{X})$ whose measures are absolutely continuous w.r.t. the Lebesgue measure. All probability measures we considered thereinafter are assumed to belong to $\mathcal{P}_2^a(\mathcal{X})$. To ease the notation, we use probability density functions such as $q(\mathbf{x}), p(\mathbf{x}), \mathbf{x} \in \mathcal{X}$ to express probability distributions in $\mathcal{P}_2^a(\mathcal{X})$. Let $(\mathcal{P}_2^a(\mathcal{X}), W_2)$ denote the metric space $\mathcal{P}_2^a(\mathcal{X})$ endowed with the 2-Wasserstein distance $W_2$, which is referred to as the quadratic Wasserstein space. We use $\nabla$ and Div to denote the gradient operator and the divergence operator, respectively.

## 2 PROBLEM FORMULATION

Consider an unnormalized probability density function $u : \mathcal{X} \to [0, \infty)$, where $\mathcal{X} \subseteq \mathbb{R}^d$ is the support of $u$. Suppose $u$ has an intractable normalizing constant $Z = \int_{\mathcal{X}} u(\mathbf{x})\mathrm{d}\mathbf{x} < \infty$. Our goal is to generate random samples from the underlying distribution $p \in \mathcal{P}_2^a(\mathcal{X})$, whose probability density function is only known up to proportionality, i.e., $p(\mathbf{x}) = u(\mathbf{x})/Z, \mathbf{x} \in \mathcal{X}$. The basic idea is to gradually optimize samples from a given distribution $q \in \mathcal{P}_2^a(\mathcal{X})$ to approximate samples from $p$, where it is easy to sample from $q$. Optimizing samples leads to functional optimization of distributions. We then introduce the classical relative entropy as the functional optimization objective. The relative entropy, a.k.a., the Kullback–Leibler divergence, for $q, p \in \mathcal{P}_2^a(\mathcal{X})$ is the average logarithmic density ratio, which is defined as

$$\mathbb{D}_{\mathrm{re}}(q\|p) = \int_{\mathcal{X}} q(\mathbf{x}) \log\left(\frac{q(\mathbf{x})}{p(\mathbf{x})}\right) \mathrm{d}\mathbf{x}. \tag{1}$$

It holds that $\mathbb{D}_{\mathrm{re}}(q\|p) \geq 0$ and $\mathbb{D}_{\mathrm{re}}(q\|p) = 0$ iff $q(\mathbf{x}) = p(\mathbf{x})$ $a.e.$ $\mathbf{x} \in \mathcal{X}$. Moreover, we denote the relative entropy functional as

$$\mathcal{F}[\cdot] := \mathbb{D}_{\mathrm{re}}(\cdot\|p) : \mathcal{P}_2^a(\mathcal{X}) \to [0, \infty]. \tag{2}$$

To sample from the unnormalized density $u = pZ$, we consider the functional minimization problem

$$\min_{q \in \mathcal{P}_2^a(\mathcal{X})} \mathcal{F}[q], \tag{3}$$

where $\mathcal{F}[q]$ is always minimized at the underlying target distribution $p$, i.e., $q(\mathbf{x}) = p(\mathbf{x})$ $a.e.$ $\mathbf{x} \in \mathcal{X}$. In a nutshell, problem (3) is an energy functional minimization problem in a metric space. To minimize the energy functional $\mathcal{F}$, it suffices to move along the corresponding gradient flow in a metric space until the flow converges. For example, a gradient flow in the Euclidean space refers to a curve whose tangent space contains the steepest descent direction of a given function. Analogously, a gradient flow in the space of probability measures means a curve that points in the steepest descent direction of a given energy functional. When equipped with the 2-Wasserstein distance, minimization of the energy functional $\mathcal{F}$ naturally corresponds to a continuous path on the quadratic Wasserstein space of distributions, which is commonly known as a Wasserstein gradient flow of the relative entropy. We call this flow a relative entropy gradient flow for briefness.

## 3 RELATIVE ENTROPY GRADIENT FLOW

In this section, we briefly review the formulation of relative entropy gradient flow and its connections to differential equations. We consider the properties of gradient flows in the quadratic Wasserstein space $(\mathcal{P}_2^a(\mathcal{X}), W_2)$. Recall that $\mathcal{F}$ in (2) is the relative entropy functional defined on $(\mathcal{P}_2^a(\mathcal{X}), W_2)$. One can show that a curve $\{q_t\}_{t\geq 0}$ in $(\mathcal{P}_2^a(\mathcal{X}), W_2)$ is a relative entropy gradient flow of $\mathcal{F}$ if it satisfies the continuity equation (Ambrosio et al. (2008), page 295 and Villani (2008), page 631),

$$\partial_t q_t = \text{Div}\left(q_t \nabla \frac{\delta \mathcal{F}[q_t]}{\delta q_t}\right), \tag{4}$$

where $q_t(\mathbf{x}) = q(t, \mathbf{x})$ evolves over time, $\frac{\delta \mathcal{F}[q_t]}{\delta q_t} = \log \frac{q_t}{p}$ is the first variation of the energy functional $\mathcal{F}$ at $q_t$, and $\nabla \frac{\delta \mathcal{F}[q_t]}{\delta q_t}$ is the Euclidean gradient of $\frac{\delta \mathcal{F}[q_t]}{\delta q_t}$. Here, we identify the gradient as the relative entropy gradient, which is defined by

$$\nabla_{W_2}\mathcal{F}[q_t] := \nabla \frac{\delta \mathcal{F}[q_t]}{\delta q_t} = \nabla \log \frac{q_t}{p}. \tag{5}$$

Moreover, the relative entropy $\mathcal{F}$ dissipates along the relative entropy gradient flow $\{q_t\}_{t\geq 0}$ at the rate (Ambrosio et al. (2008), page 295)

$$\partial_t \mathcal{F}[q_t] = -\mathbb{E}_{q_t}[\|\nabla_{W_2}\mathcal{F}[q_t]\|^2]. \tag{6}$$

Therefore, the relative entropy gradient flow $\{q_t\}_{t\geq 0}$ eventually converges to the target distribution $p$ as $t \to \infty$. As pointed out in Ambrosio et al. (2008) (Page 175), under mild conditions the continuity equation (4) concerning $\{q_t\}_{t\geq 0}$ determines a time-inhomogeneous Markov process $\{X_t\}_{t\geq 0}$ that starts at a random particle $X_0 \sim q_0$ and follows the particle evolution dynamics

$$\frac{dX_t}{dt} = \mathbf{v}_t(X_t), \ X_t \sim q_t, \ t \geq 0. \tag{7}$$

Note that the velocity fields

$$\mathbf{v}_t = -\nabla_{W_2}\mathcal{F}[q_t] = \nabla \log \frac{p}{q_t}, \ t \geq 0 \tag{8}$$

drive the evolution of the particle $X_t$ in the Euclidian space, which results in the transport of $q_t$ in $(\mathcal{P}_2^a(\mathcal{X}), W_2)$. An important observation is that

$$\mathbf{v}_t = \nabla \log \frac{p}{q_t} = \nabla \log \frac{u}{q_t}, \ t \geq 0. \tag{9}$$

Therefore, the velocity fields do not involve the unknown normalizing constant $Z$. This is the key motivation for us to use the relative entropy gradient flow in the proposed method.

## 4 SAMPLING AS PARTICLE EVOLUTION

As indicated by the energy dissipation of relative entropy $\mathcal{F}$ in (6), running the relative entropy gradient flow $\{q_t\}_{t\geq 0}$ dynamics can provide a nice approximate solution to the functional minimization problem (3) when time $t$ is large enough. Therefore, to sample from the target distribution $p$, it is appropriate to simulate the relative entropy gradient flow $\{q_t\}_{t\in[0,T]}$ with the time horizon $T$ sufficiently large. A natural strategy is to discretize the particle evolution form of relative entropy gradient flow in (7) with forward Euler iterations (LeVeque, 2007) as follows,

$$X_{k+1} = X_k + s\mathbf{v}_k(X_k), \ X_0 \sim q_0, \ k = 0, 1, \ldots, K-1, \tag{10}$$

with the velocity field at step $k$

$$\mathbf{v}_k = -\nabla_{W_2}\mathcal{F}[q_k] = \nabla \log \frac{p}{q_k}, \tag{11}$$

where $s > 0$ is a tunable small step size, $K = \lfloor T/s \rfloor$ is the number of iterations and $q_k$ is the corresponding discretized gradient flow at step $k$, i.e., $X_k \sim q_k$. Combining the expressions in (10) and (11), we have that the iterations progress according to

$$X_{k+1} = X_k + s\nabla \log \frac{p}{q_k}, \ X_0 \sim q_0, \ k = 0, 1, \ldots, K-1. \tag{12}$$

In principle, it is necessary to evaluate the velocity field $\mathbf{v}_k = \nabla \log(p/q_k)$ each iteration in (12). By (9), the velocity field of the relative entropy gradient flow can be simplified to

$$\mathbf{v}_k = \nabla \log \frac{p}{q_k} = \nabla \log \frac{u}{q_k}, \ k = 0, 1, \ldots, K-1, \tag{13}$$

where $u = Zp$ is the given unnormalized density of the target distribution $p$. Then only the density $q_k$ remains unknown for evaluating the velocity field. Ideally, $q_k$ can be estimated by evolving a large number of particles $\{X_k^i\}_{i=1}^N$. However, direct estimation of $q_k$ is difficult due to the curse of dimensionality and the potential expensive computation cost for different $k$s. Our solution is to approximate the velocity field (13) as a whole.

Assuming a nice approximation $\widehat{\mathbf{v}}_k$ of the velocity field (13) is provided, then one can implement the following iterations for approximately sampling from $q_K$ with no effort,

$$\widetilde{X}_{k+1} = \widetilde{X}_k + s\widehat{\mathbf{v}}_k(\widetilde{X}_k), \ \widetilde{X}_0 \sim q_0, \ k = 0, 1, \ldots, K-1. \tag{14}$$

Through the iterations above, we can collect $\widetilde{X}_k \sim \tilde{q}_k \approx q_k, \ k = 1, 2, \ldots, K$. We will discuss approximation of the velocity field $\mathbf{v}_k = \nabla \log(u/q_k)$ from the perspective of estimating the logarithmic density ratio $\log(u/q_k)$ in the next section.

## 5  LOGARITHMIC DENSITY RATIO ESTIMATION AND THE RELATIVE ENTROPY GRADIENT SAMPLER

In this section, we first propose a novel estimation procedure of the logarithmic density ratio $\log(u/q)$ based on an unnormalized density $u$ and random samples from $q$.

We use a model ratio $R : \mathcal{X} \to [0, \infty)$ to fit the true ratio $R_{uq}^\star = u/q$ between a density $q$ and an unnormalized density $u$. Let $g : \mathbb{R} \to \mathbb{R}$ be a differentiable and strictly convex function. A Bregman score (Dawid, 2007; Gneiting & Raftery, 2007; Kanamori & Sugiyama, 2014) with the base probability measure $q \in \mathcal{P}_2^a(\mathcal{X})$ to measure the discrepancy between $R$ and $R_{uq}^\star$ is defined by

$$\mathfrak{B}(R) = \mathbb{E}_{X \sim q}[g'(R(X))R(X) - g(R(X))] - \mathbb{E}_{X \sim w}\left[\frac{u(X)}{w(X)}g'(R(X))\right],$$

where $w \in \mathcal{P}_2^a(\mathcal{X})$ is an introduced and reference distribution for calculating the integral involving $u$. It should be easy to sample from $w$ and the support of $u$ should be included in the support of $w$. Additionally, $\mathfrak{B}(R) \geq \mathfrak{B}(R_{uq}^\star)$, where the equality holds iff $R(\mathbf{x}) = R_{uq}^\star(\mathbf{x})$ $(q, u)$-a.e. $\mathbf{x} \in \mathcal{X}$.

In this work, we take $g(x) = x \log(x) - x$. We use this function for two reasons: (a) convexity, this is to satisfy the basic requirement of the Bregman score; (b) cancellation of the unknown normalizing constant $Z$ of $u$. Simple calculation shows that $\mathfrak{B}(R)$ can be written as

$$\mathfrak{B}(R) = \mathbb{E}_{X \sim q}[R(X)] - \mathbb{E}_{X \sim w}\left[\frac{u(X)}{w(X)}\log(R(X))\right]. \tag{15}$$

Recall that the true density ratio $R_{uq}^\star$ can be factorized as $R_{uq}^\star = u/q = Z(p/q)$. Thus the numerical scale of the true density ratio $R_{uq}^\star$ hinges on two factors, i.e., the normalizing constant $Z$ of $u$ and the standard density ratio $p/q$. Since numerical scales of these factors are difficult to determine in applications, the induced numerical instability can deteriorate the density ratio estimate. In order to prevent the density ratio estimation from such instability, we consider the model ratio $R$ on the logarithmic scale. This will also release the nonnegative constraint on $R$ as a byproduct.

From now on, we denote $D_{uq}^\star = \log(R_{uq}^\star)$, $D = \log(R) : \mathcal{X} \to \mathbb{R}$. Then $\mathfrak{B}(D)$ can be rewritten as

$$\mathfrak{B}(D) = \mathbb{E}_{X \sim q}[\exp(D(X))] - \mathbb{E}_{X \sim w}\left[\frac{u(X)}{w(X)}D(X)\right]. \tag{16}$$

It can be shown that the logarithmic density ratio $D_{uq}^\star$ is identifiable at the population level by minimizing (16) with respect to $D$.

**Theorem 1.** *For $\mathfrak{B}(D)$ defined in (16), we have $D_{uq}^\star \in \arg\min_D \mathfrak{B}(D)$. In addition, for any $D$ with $\mathbb{E}_{X \sim w}\left[\frac{u(X)}{w(X)}D(X)\right] < \infty$, $\mathfrak{B}(D) \geq \mathfrak{B}(D_{uq}^\star)$, with equality iff $D(\mathbf{x}) = D_{uq}^\star(\mathbf{x})$ $(q, u)$-a.e. $\mathbf{x} \in \mathcal{X}$.*

---

**Algorithm 1:** REGS: Relative entropy gradient sampler

---

> **Input**: $u = Zp$                                 `// unnormalized target density`
> step size $s > 0$, an integer $K > 0$,            `// step size, maximum loop count`
> $\widetilde{X}_0^i \sim q_0,\ i = 1, 2, \ldots, n$                  `// initial particles`
> $w \in \mathcal{P}_2^a(\mathcal{X})$                           `// reference distribution`
> $k \leftarrow 0$
> **while** $k < K$ **do**
> > $Y_k^i \sim w,\ i = 1, 2, \ldots, n$                `// reference samples`
> > $\widehat{D}_{\phi_k} \in \arg\min_{D_\phi} \frac{1}{n} \sum_{i=1}^n \left[ \exp(D_\phi(\widetilde{X}_k^i)) - \frac{u(Y_k^i)}{w(Y_k^i)} D_\phi(Y_k^i) \right]$    `// log density ratio`
> > $\widehat{\mathbf{v}}_k(\mathbf{x}) = \nabla \widehat{D}_{\phi_k}(\mathbf{x})$                   `// velocity field`
> > $\widetilde{X}_{k+1}^i = \widetilde{X}_k^i + s\widehat{\mathbf{v}}_k(\widetilde{X}_k^i), i = 1, 2, \ldots, n$      `// update particles`
> > $k \leftarrow k + 1$
> **end**
> **Output**: $\widetilde{X}_K^i \sim \tilde{q}_K \approx p, i = 1, 2, \ldots, n$               `// output particles`

---

Based on Theorem 1, we can estimate the unknown logarithmic density ratio $D_{uq_k}^\star = \log(u/q_k)$ with a deep neural network $D_\phi$ with parameter $\phi$ through the sample version of (16). Let $\{\widetilde{X}_k^i\}_{i=1}^n$ be i.i.d. samples from $\tilde{q}_k \approx q_k$ and $\{Y_k^i\}_{i=1}^n$ be i.i.d. samples from a reference distribution $w$. We solve the following deep nonparametric estimation problem via stochastic gradient descent (SGD) for $\widehat{D}_{\phi_k}$

$$\widehat{D}_{\phi_k} \in \arg\min_{D_\phi} \widehat{\mathfrak{B}}(D_\phi) = \frac{1}{n} \sum_{i=1}^n \left[ \exp(D_\phi(\widetilde{X}_k^i)) - \frac{u(Y_k^i)}{w(Y_k^i)} D_\phi(Y_k^i) \right]. \tag{17}$$

With the logarithmic density ratio estimator $\widehat{D}_{\phi_k}$, the velocity field $\mathbf{v}_k$ in (13) can be approximately computed by $\widehat{\mathbf{v}}_k = \nabla \widehat{D}_{\phi_k}$. By considering sampling as a particle evolution process discussed in Section 4, REGS updates the initial particles $\{\widetilde{X}_0^i\}_{i=1}^n$ with iterations in (14) as follows:

$$\widetilde{X}_{k+1}^i = \widetilde{X}_k^i + s\widehat{\mathbf{v}}_k(\widetilde{X}_k^i), \ \ \widetilde{X}_0^i \sim q_0, \ \ i = 1, 2, \ldots, n, \ \ k = 0, 1, \ldots, K - 1. \tag{18}$$

We summarize the proposed REGS for sampling from an unnormalized density in Algorithm 1.

## 6   Numerical experiments

We evaluate REGS on a large number of 1D and 2D mixture distributions and test its stability in the high-dimensional setting with multivariate Gaussian distributions. We also use REGS to perform Bayesian logistic regression on benchmark datasets. For comparison, we consider three existing methods including SVGD (Liu & Wang, 2016), ULA (Roberts & Tweedie, 1996) and MALA (Roberts & Tweedie, 1996). All experiments are done using a NVIDIA Tesla K80 GPU and common CPU computing resources. The neural network architecture, hyperparameter values, dataset descriptions, and additional experimental results are given in the appendix. The python code of REGS is available at `https://github.com/anonymous/REGS`.

### 6.1   Mixture distributions

We run REGS and SVGD, ULA and MALA to generate 2000 particles for mixtures of 2, 8 and 9 Gaussians (see Scenarios 4, 5, 6 in Appendix B), and 5000 particles for a mixture of 25 Gaussians (see Scenario 9 in Appendix B). The sampling qualities of these algorithms are compared by scatter plots with density contours of target mixture distributions. We classify all scatter points with labels according to the nearest mode, and plot the histograms of the label counts.

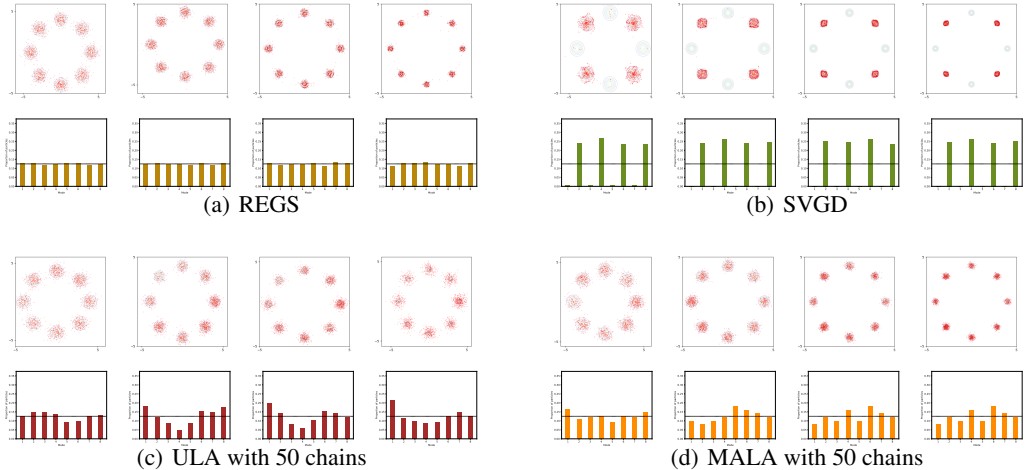

Figure 2: Mixtures of 8 Gaussians with *equal weights*: scatter plots and histograms of generated samples by (a) REGS, (b) SVGD, (c) ULA with 50 chains, and (d) MALA with 50 chains. From left to right in each subfigure, the variance of Gaussians varies from $\sigma^2 = 0.2$ (first column), $\sigma^2 = 0.1$ (second column), $\sigma^2 = 0.05$ (third column), to $\sigma^2 = 0.03$ (fourth column).

**Gaussian mixtures with equal weights**  Figure 2 shows the scatter plots and histograms of samples generated by (a) REGS, (b) SVGD, (c) ULA with 50 chains, and (d) MALA with 50 chains from mixtures of 8 Gaussians with *equal weights*. It shows that REGS is able to explore all the components in the mixture distribution nearly equally. However, SVGD is only able to find part of the modes, as indicated in Figures 2(b). Figures 2(c) and 2(d) show that MALA and ULA with 50 chains find all modes but with unequal weights, especially as the variance of each component decreases.

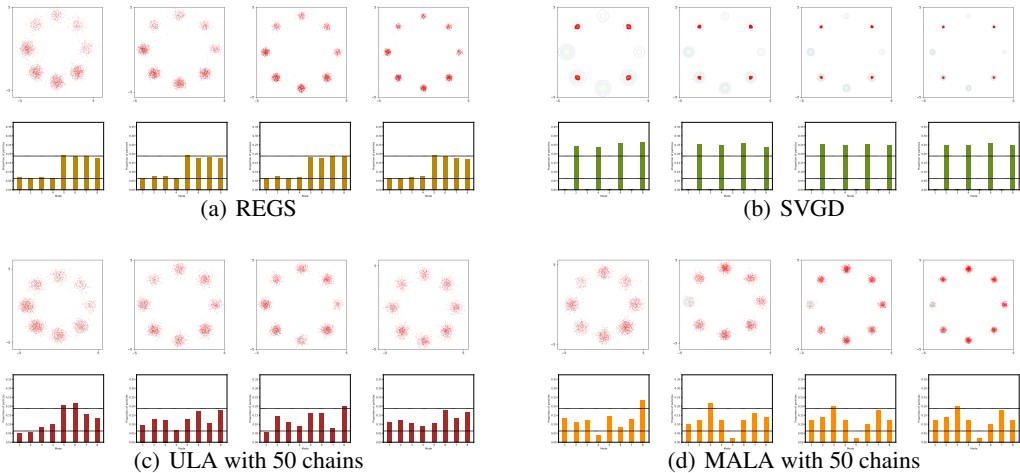

Figure 3: Mixtures of 8 Gaussians with *unequal weights*: scatter plots and histograms of generated samples by (a) REGS, (b) SVGD, (c) ULA with 50 chains, and (d) MALA with 50 chains. From left to right in each subfigure, the variance of Gaussians varies from $\sigma^2 = 0.2$ (first column), $\sigma^2 = 0.1$ (second column), $\sigma^2 = 0.05$ (third column), to $\sigma^2 = 0.03$ (fourth column).

**Gaussian mixtures with unequal weights**  Figure 3 shows the scatter plots and histograms of samples generated by (a) REGS, (b) SVGD, (c) ULA with 50 chains, and (d) MALA with 50 chains from mixtures of 8 Gaussians with *unequal weights* $(1, 1, 1, 1, 3, 3, 3, 3)/16$. Figure 3(a) shows that the samples generated by REGS have the correct weights. Figures 3(c) and 3(d) indicate that ULA

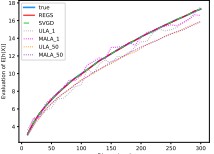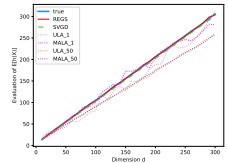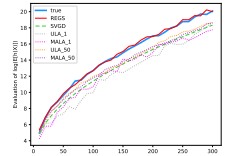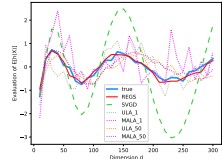

Figure 4: Monte Carlo estimates of $\mathbb{E}[h(X)]$ versus $d$ for $d$-dimensional multivariate Gaussian distributions of $X$, For $d$ increasing from 10 to 300 with lag 10. From left to right, $h(x) = \alpha^\mathrm{T} x$, $(\alpha^\mathrm{T} x)^2$, $\exp(\alpha^\mathrm{T} x)$, and $10\cos(\alpha^\mathrm{T} x + 1/2)$ with $\alpha \in \mathbb{R}^d, \|\alpha\|_2 = 1$. The curves represent the estimates using the target samples ("true", blue solid line) and the generated samples by REGS (red solid line), SVGD (green dash line), ULA_1: gray dotted line, MALA_1: pink dotted line, ULA_50: oringe dotted line, and MALA_50: orchid dotted line.

and MALA assign particles to modes with incorrect weights. Moreover, the quality of the samples generated by SVGD, ULA and MALA deteriorates as the number of modes increases, while the performance of REGS remains stable. We also included the results from ULA and MALA with a single chain in Figures 7 and 10 in Appendix D, which show that these samplers have difficulty with multimodal distributions if only a single chain is used.

To further analyze the performance, we report the Monte Carlo estimates of $\mathbb{E}[h(X)]$ using a test function $h$ in Table 1, where $h(x) = \alpha^\mathrm{T} x$, $(\alpha^\mathrm{T} x)^2$, and $10\cos(\alpha^\mathrm{T} x + 1/2)$ with $\alpha \in \mathbb{R}^2, \|\alpha\|_2 = 1$, and $X$ is distributed as various Gaussian mixtures with unequal weights. By comparing the Monte Carlo estimates of $\mathbb{E}[h(X)]$ using the samplers with the values based on target samples, we see that REGS performs better and is more stable than SVGD, ULA and MALA, especially when $h(x) = 10\cos(\alpha^\mathrm{T} x + 1/2)$. We include additional numerical results including more scatter plots and histograms (Figure 6-10) and Monte Carlo estimates with equal wieghts (Table 6) in Appendix D.

Table 1: Monte Carlo estimates of $\mathbb{E}[h(X)]$ with four samplers for 2D mixtures of Gaussians random vectors $X$ with unequal weights. "Target" denotes the Monte Carlo estimate with target samples. ULA_$k$ and MALA_$k$ denote the ULA and MALA with $k$ chains, respectively.

| Distributions | $\sigma^2$ | $h(x) = \alpha^\mathrm{T} x$ | | | | | | | $h(x) = (\alpha^\mathrm{T} x)^2$ | | | | | | | $h(x) = 10\cos(\alpha^\mathrm{T} x + 1/2)$ | | | | | | |
|---|---|---|---|---|---|---|---|---|---|---|---|---|---|---|---|---|---|---|---|---|---|---|
| | | Target | REGS | SVGD | ULA_1 | MALA_1 | ULA_50 | MALA_50 | Target | REGS | SVGD | ULA_1 | MALA_1 | ULA_50 | MALA_50 | Target | REGS | SVGD | ULA_1 | MALA_1 | ULA_50 | MALA_50 |
| 2gaussian | 0.2 | -0.71 | **-0.61** | -0.05 | -2.86 | -2.85 | 0.46 | 0.00 | 2.20 | **2.20** | 32.40 | 8.39 | 8.35 | 8.16 | 8.24 | 3.39 | **3.08** | -7.92 | -6.42 | -6.29 | -7.73 | -7.43 |
| | 0.1 | -0.71 | **-0.47** | -0.07 | -2.83 | -2.82 | 0.45 | 0.00 | 2.12 | **2.10** | 32.20 | 8.11 | 8.09 | 8.10 | 8.13 | 3.49 | **2.80** | -8.11 | -6.54 | -6.39 | -8.10 | -7.81 |
| | 0.05 | -0.71 | **-0.48** | -0.03 | -2.84 | -2.84 | 0.45 | -0.00 | 2.07 | **2.05** | 32.10 | 8.10 | 8.15 | 8.05 | 8.10 | 3.58 | **2.91** | -8.16 | -6.75 | -6.60 | -8.32 | -7.94 |
| | 0.03 | -0.70 | **-0.52** | 0.03 | -2.82 | -2.83 | 0.45 | - | 2.03 | **2.03** | 31.90 | 7.98 | 8.18 | 8.04 | - | 3.69 | **3.08** | -8.25 | -6.70 | -6.29 | -8.40 | - |
| 8gaussian | 0.2 | -1.20 | **-1.20** | -0.06 | -0.49 | -1.72 | 0.09 | -1.30 | 8.23 | **8.20** | 8.05 | 9.93 | 8.63 | 7.56 | 8.54 | -3.16 | **-3.16** | 1.46 | -5.24 | -5.70 | -2.71 | -3.48 |
| | 0.1 | -1.21 | **-1.15** | -0.02 | 0.00 | -0.68 | 0.40 | -0.22 | 8.11 | **8.08** | 8.30 | 0.10 | 2.08 | 7.94 | 8.63 | -3.31 | **-3.30** | 1.33 | 8.35 | 4.63 | -3.30 | -3.29 |
| | 0.05 | -1.21 | **-1.12** | -0.01 | 0.00 | -2.83 | 0.50 | -0.27 | 8.06 | **8.01** | 8.09 | 0.05 | 8.09 | 8.05 | 8.24 | -3.41 | **-3.35** | 1.45 | 8.54 | -6.53 | -3.56 | -2.43 |
| | 0.03 | -1.21 | **-1.12** | -0.03 | 0.00 | -2.66 | 0.50 | -0.28 | 8.05 | **8.00** | 8.10 | 0.03 | 7.95 | 8.03 | 8.55 | -3.46 | **-3.40** | 1.41 | 8.64 | -5.22 | -3.59 | -3.14 |
| 25gaussian | 0.2 | 1.00 | **1.00** | 1.64 | 1.17 | 0.94 | 0.90 | 0.92 | 8.05 | **8.04** | 9.43 | 68.02 | 48.48 | 7.62 | 7.88 | 0.21 | 0.17 | 0.12 | 0.74 | 0.33 | 0.27 | **0.22** |
| | 0.1 | 1.00 | **1.00** | 0.04 | 2.11 | 0.91 | 0.98 | 0.85 | 7.97 | **7.94** | 2.04 | 53.03 | 51.55 | 7.29 | 7.79 | 0.18 | **0.18** | 3.56 | -1.41 | -0.28 | 0.52 | 0.33 |
| | 0.05 | 1.00 | **0.91** | 0.07 | 1.42 | 1.16 | -0.03 | 0.46 | 7.90 | **7.83** | 1.07 | 13.69 | 47.61 | 4.79 | 7.82 | 0.19 | **0.17** | 5.07 | -3.30 | -0.08 | 0.53 | 0.41 |
| | 0.03 | 1.00 | **0.81** | -0.02 | 0.00 | 0.27 | -0.11 | 0.27 | 7.87 | **7.70** | 0.96 | 0.20 | 53.18 | 4.75 | 7.43 | 0.17 | **0.16** | 5.68 | 8.64 | -0.02 | 0.08 | -0.01 |

## 6.2 Multivariate Gaussian distribution

Let the target distribution be a $d$-dimensional Gaussian distribution with mean $\mu = (1, 1, \cdots, 1) \in \mathbb{R}^d$ and covariance matrix $\Sigma \in \mathbb{R}^{d \times d}, \Sigma_{i,j} = \rho^{|i-j|}$ with $\rho = 0.7$. We consider four test functions $h(x)$, i.e., $h(x) = \alpha^\mathrm{T} x$ (the first moment), $h(x) = (\alpha^\mathrm{T} x)^2$ (the second moment), $h(x) = \exp(\alpha^\mathrm{T} x)$ (the moment generating function), and $h(x) = 10\cos(\alpha^\mathrm{T} x + 1/2)$ with $\alpha \in \mathbb{R}^d$ satisfying $\|\alpha\|_2 = 1$. For reference, we provide the Monte Carlo estimates of $\mathbb{E}[h(X)]$ using target samples. We compare REGS with SVGD, ULA_1, MALA_1, ULA_50, MALA_50 in Figure 4, the number of particles is 5000 for each sampler, where ULA_$k$ and MALA_$k$ denote the ULA and MALA with $k$ chians. For ULA and MALA, because of large variations of the estimates, we repeat the process 10 times and compute the average as the final estimate. Figure 4 presents these Monte Carlo estimates as $d$ increases from 10 to 300 with step size 10. The logarithm of the estimated $\mathbb{E}[\exp(\alpha^\mathrm{T} X)]$ is shown. As shown in Figure 4, the estimates using REGS and SVGD have smaller fluctuations than those using ULA and MALA, although all four methods can estimate $\mathbb{E}[\alpha^\mathrm{T} X]$ and $\mathbb{E}[(\alpha^\mathrm{T} X)^2]$ well. Moreover, the third and the fourth panels in Figure 4 show that REGS outperforms SVGD, ULA and MALA when $h(x) = \exp(\alpha^\mathrm{T} x)$ or $10\cos(\alpha^\mathrm{T} x + 1/2)$.

## 6.3 BAYESIAN LOGISTIC REGRESSION

We apply REGS to Bayesian logistic regression for binary classification on five datasets, including *Banana*, *German*, *Image*, *Ringnorm*, and *Covertype*. These datasets were analyzed in Liu & Wang (2016) and the first four datasets had been analyzed in Gershman et al. (2012). We consider a similar setting to that in (Liu & Wang, 2016; Gershman et al., 2012), which assigns a Gaussian prior $\pi(\boldsymbol{\beta}|\alpha) = \mathcal{N}(\mathbf{0}, \alpha^{-1}\mathbf{I})$ to the regression coefficient $\boldsymbol{\beta}$ (including the intercept). We specify the prior of $\alpha$ as $\pi(\alpha) = \text{Gamma}(1, 0.01)$. For comparison, we consider SVGD, ULA and MALA. The inference is based on the posterior $\pi(\boldsymbol{\beta}|\text{data})$.

These datasets are partitioned randomly into two parts, the training sets (80%) and the test sets (20%). We repeats the random partition 10 times. We evaluate the classification accuracy on test data with 5000 particles from the posterior. Table 2 lists the averages and standard errors (in parenhteses) of test accuracy. From Table 2 we can see that REGS is comparable with SVGD, ULA and MALA. For the *Covertype* dataset, MALA failed to converge, so no results from it are included in Table 2.

Table 2: Averages and standard errors (in parenhteses) of classification accuracy on test data from five datasets, $d$: number of features, $N$: sample size.

| datasets | $d$ | $N$ | Averages of Accuracy (%) | | | | | |
|---|---|---|---|---|---|---|---|---|
| | | | REGS | SVGD | ULA_1 | MALA_1 | ULA_50 | MALA_50 |
| Banana | 2 | 5300 | 54.1 (3.1) | **55.5** (2.9) | 55.1 (1.9) | 55.2 (1.9) | 55.1 (1.9) | 55.2 (1.9) |
| German | 20 | 1000 | **77.2** (2.2) | 75.6 (1.2) | 76.5 (1.8) | 76.6 (2.2) | 76.6 (2.0) | 76.6 (2.1) |
| Image | 18 | 2086 | **83.4** (1.5) | 82.8 (1.7) | 82.7 (2.3) | 82.9 (2.3) | 82.8 (2.3) | 82.8 (2.3) |
| Ringnorm | 20 | 7400 | **76.3** (0.9) | 75.9 (1.0) | 75.7 (1.4) | 75.7 (1.4) | 75.7 (1.4) | 75.2 (1.4) |
| Covertype | 54 | 581012 | 75.0 (1.2) | **75.6** (0.8) | 74.1 (0.3) | – | 74.2 (0.4) | – |

## 6.4 DISCUSSION OF THE EXPERIMENTAL RESULTS

The experimental results reported above and in the appendix indicate that REGS is capable of generating better quality samples than SVGD, ULA and MALA from Gaussian mixture distributions. Also, the results suggest that particles generated by REGS can cross valleys in the landscape of a multimodal distribution even if they are initialized in a different regions. An intuitive explanation is as follows. The movement of the REGS particles is determined by the velocity field. If the velocity field is not zero at a particle, the particle will continue to evolve towards the target distribution. Moreover, all particles interact with each other through the velocity field, which is beneficial in sampling from multimodal distributions. For ULA and MALA, there are no interactions among particles or incentives for particles to cross valleys between two modes, thus it is more difficult for these methods to sample from multimodal distributions. A possible remedy is to use multiple chains as we did in the above experiments. To some extend, this alleviates the problem encountered in sampling from multimodal distributions. However, the success of this strategy depends on the initial samples being near the modes as well as having the correct proportions of the initial samples being close to each mode. In comparison, REGS uses a principled way to move particles from an initial reference distribution to a multimodal distribution, albeit with a higher computational cost.

## 7 CONCLUSION

We have introduced REGS, a novel gradient flow based method for sampling from unnormalized distributions. Extensive numerical experiments demonstrate that REGS performs better than several existing popular sampling methods in the setting of challenging multimodal mixture distributions. In future work, we hope to establish the convergence properties of REGS generated sampling distributions as the numbers of iterations and particles increase.

As with any sampling algorithms, there is a trade-off between sampling quality and computational efficiency. On one hand, as our numerical experiments demonstrate, REGS can generate samples with better quality than the three existing methods we considered in the challenging mixture model settings. On the other hand, REGS is computationally more expensive, as it involves neural network training in the iterations, compared with existing methods such as ULA and MALA that can be implemented more quickly. As computational power continues to increase rapidly, REGS can be a useful addition to the toolkit of sampling methods for multimodal distributions.

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

## A   APPENDIX

In this appendix, we prove Theorem 1, give a detailed description of the models in the numerical experiments, the neural network architecture used in implementing REGS, and additional numerical results.

## B   PROOF OF THEOREM 1

**Proof**. By the definition of Bregman score $\mathfrak{B}(R)$ between $u$ and $q$, it is easy to check

$$R_{uq}^{\star} \in \arg\min_{R} \mathfrak{B}(R),$$

$\mathfrak{B}(R) \geq \mathfrak{B}(R_{uq}^{\star})$ with equality iff $R(\mathbf{x}) = R_{uq}^{\star}(\mathbf{x})$ $(q, u)$-a.e. $\mathbf{x} \in \mathcal{X}$.

Since $D_{uq}^{\star} = \log(R_{uq}^{\star}), D = \log(R), \mathfrak{B}(R) = \mathfrak{B}(D)$, when $\mathbb{E}_{X \sim w} \left[ \frac{u(X)}{w(X)} D(X) \right] < \infty$, we have

$$D_{uq}^{\star} \in \arg\min_{D} \mathfrak{B}(D),$$

$\mathfrak{B}(D) \geq \mathfrak{B}(D_{uq}^{\star})$ with equality iff $D(\mathbf{x}) = D_{uq}^{\star}(\mathbf{x})$ $(q, u)$-a.e. $\mathbf{x} \in \mathcal{X}$.   ∎

## C   GAUSSIAN MIXTURE DISTRIBUTIONS

Let the density of a $d$-dimensional multivariate Gaussian distribution with mean $\mu \in \mathbb{R}^d$ and covariance matrix $\Sigma \in \mathbb{R}^{d \times d}$ be

$$f(x; \mu, \Sigma) = \frac{1}{(2\pi)^{\frac{d}{2}} \sqrt{\det(\Sigma)}} \exp\left\{ -\frac{1}{2}(x - \mu)^{\mathrm{T}} \Sigma^{-1}(x - \mu) \right\}.$$

We consider several scenarios below for the unnormalized density function $u(x)$.

Scenario 1. *2Gaussians_1d1*. One-dimensional mixture of 2 Gaussians,

$$u(x) = \frac{1}{3} f(x; \mu_1, \sigma_1^2) + \frac{2}{3} f(x; \mu_2, \sigma_2^2),$$

where $\mu_1 = 1, \mu_2 = -2$, and $\sigma_1^2 = 0.25, \sigma_2^2 = 2$;

Scenario 2. *2Gaussians_1d2*. One-dimensional mixture of 2 Gaussians,

$$u(x) = \frac{1}{3} f(x; \mu_1, \sigma_1^2) + \frac{2}{3} f(x; \mu_2, \sigma_2^2),$$

where $\mu_1 = 3, \mu_2 = -3$, and $\sigma_1^2 = 0.25, \sigma_2^2 = 2$;

Scenario 3. *2Gaussians_1d3*. One-dimensional mixture of 2 Gaussians,

$$u(x) = \frac{1}{3} f(x; \mu_1, \sigma_1^2) + \frac{2}{3} f(x; \mu_2, \sigma_2^2),$$

where $\mu_1 = 3, \mu_2 = -3$, and $\sigma_1^2 = 0.03, \sigma_2^2 = 0.03$;

Scenario 4. *2Gaussians*. Two-dimensional mixture of 2 Gaussians,

$$u(x) = f(x; \mu_1, \Sigma_1) + f(x; \mu_2, \Sigma_2),$$

where $\mu_1 = (r, 0)^{\mathrm{T}}, \mu_2 = (-r, 0)^{\mathrm{T}}$, and $\Sigma_1 = \Sigma_2 = \sigma^2 \mathbf{I}$.

Scenario 5. *8Gaussians*. Two-dimensional mixture of 8 Gaussians,

$$u(x) = \sum_{j=1}^{8} f(x; \mu_j, \Sigma_j),$$

where $\mu_j = r(\sin(2(j-1)\pi/8), \cos(2(j-1)\pi/8))^{\mathrm{T}}$, and $\Sigma_j = \sigma^2 \mathbf{I}$ for $j = 1, \cdots, 8$.

**Scenario 6.** *9Gaussians*. Two-dimensional mixture of 9 Gaussians,

$$u(x) = \sum_{j=1}^{3}\sum_{k=1}^{3} f(x; \mu_{jk}, \Sigma_{jk}),$$

where $\mu_{jk} = 4(j-2, k-2)^{\mathrm{T}}$, $\Sigma_{jk} = \sigma^2 \mathbf{I}$ for $j = 1, 2, 3$ and $k = 1, 2, 3$.

**Scenario 7.** *16Gaussians_1c*. Two-dimensional mixture of 16 Gaussians,

$$u(x) = \sum_{j=1}^{16} f(x; \mu_j, \Sigma_j),$$

where $\mu_j = 4(\sin(2(j-1)\pi/16), \cos(2(j-1)\pi/16))^{\mathrm{T}}$, and $\Sigma_j = 0.03\mathbf{I}$ for $j = 1, \cdots, 16$.

**Scenario 8.** *16Gaussians_2c*. Two-dimensional mixture of 16 Gaussians,

$$u(x) = \sum_{j=1}^{8} f(x; \mu_j, \Sigma_j) + \sum_{k=1}^{8} f(x; \mu_k, \Sigma_k),$$

where $\mu_j = 4(\sin(2(j-1)\pi/8), \cos(2(j-1)\pi/8))^{\mathrm{T}}$, $\mu_k = 2(\sin(2(j-1)\pi/8), \cos(2(j-1)\pi/8))^{\mathrm{T}}$, and $\Sigma_j = \Sigma_k = 0.03\mathbf{I}$ for $j = 1, \cdots, 8$ and $k = 1, \cdots, 8$.

**Scenario 9.** *25Gaussians*. Two-dimensional mixture of 25 Gaussians,

$$u(x) = \sum_{j=1}^{5}\sum_{k=1}^{5} f(x; \mu_{jk}, \Sigma_{jk}),$$

where $\mu_{jk} = 2(j-3, k-3)^{\mathrm{T}}$, $\Sigma_{jk} = \sigma^2 \mathbf{I}$ for $j = 1, \cdots, 5$ and $k = 1, \cdots, 5$.

**Scenario 10.** *49Gaussians*. Two-dimensional mixture of 49 Gaussians,

$$u(x) = \sum_{j=1}^{7}\sum_{k=1}^{7} f(x; \mu_{jk}, \Sigma_{jk}),$$

where $\mu_{jk} = \frac{3}{2}(j-4, k-4)^{\mathrm{T}}$, $\Sigma_{jk} = 0.03\mathbf{I}$ for $j = 1, \cdots, 7$ and $k = 1, \cdots, 7$.

**Scenario 11.** *81Gaussians*. Two-dimensional mixture of 81 Gaussians,

$$u(x) = \sum_{j=1}^{9}\sum_{k=1}^{9} f(x; \mu_{jk}, \Sigma_{jk}),$$

where $\mu_{jk} = \frac{3}{2}(j-5, k-5)^{\mathrm{T}}$, $\Sigma_{jk} = 0.03\mathbf{I}$ for $j = 1, \cdots, 9$ and $k = 1, \cdots, 9$.

**Scenario 12.** *1circle*. Let $\mu_i = 4(\cos(2i\pi/N), \sin(2i\pi/N))^{\mathrm{T}}$, $i = 0, 1, \cdots, N-1$ with $N = 400$. Consider three noise to be added to each point $\mu_i$, including the uniform distribution $U(\mu_i, 1/30)$ on a disc with center at $\mu_i$ and radius 1/30, Gaussian distribution $N(\mu_i, 0.03\mathbf{I})$, and mixed these two distributions.

**Scenario 13.** *2circles*. Let $\mu_{1i} = 2(\cos(2i\pi/N), \sin(2i\pi/N))^{\mathrm{T}}$ and $\mu_{2i} = 4(\cos(2i\pi/N), \sin(2i\pi/N))^{\mathrm{T}}$, $i = 0, 1, \cdots, N-1$ with $N = 200$. Consider three noise to be added to each point $\mu_{ki}$ including uniform distribution $U(\mu_{ki}, 1/30)$ on a disc with center at $\mu_{ki}$ and radius 1/30, Gaussian distribution $N(\mu_{ki}, 0.03\mathbf{I})$, and mixed these two distributions, $k = 1, 2$.

**Scenario 14.** *1spiral*. Let $\mu_i = \frac{4i\pi}{N}\cos(4i\pi/N), \sin(4i\pi/N))^{\mathrm{T}}$, $i = 0, 1, \cdots, N-1$ with $N = 400$. Consider three noise to be added to each point $\mu_i$, including the uniform distribution $U(\mu_i, 1/30)$ on a disc with center at $\mu_i$ and radius 1/30, Gaussian distribution $N(\mu_i, 0.03\mathbf{I})$, and mixed these two distributions.

**Scenario 15.** *2spirals*. Let $\mu_{1i} = \frac{3i\pi}{N}(\cos(3i\pi/N), \sin(3i\pi/N))^{\mathrm{T}}$ and $\mu_{2i} = -\frac{3i\pi}{N}(\cos(3i\pi/N), \sin(3i\pi/N))^{\mathrm{T}}$, $i = 0, 1, \cdots, N-1$ with $N = 200$. Consider three noise to be added to each point $\mu_{ki}$ including uniform distribution $U(\mu_{ki}, 1/30)$ on a disc with center at $\mu_{ki}$ and radius 1/30, Gaussian distribution $N(\mu_{ki}, 0.03\mathbf{I})$, and mixed these two distributions, $k = 1, 2$.

**Scenario 16.** *moons.* Let $\mu_{1i} = (8i/N - 6, 4\sin(i\pi/N))^{\mathrm{T}}$ and $\mu_{2i} = (8i/N - 2, 4\sin(i\pi/N))^{\mathrm{T}}$, $i = 0, 1, \cdots, N - 1$ with $N = 200$. Consider three noise to be added to each point $\mu_{ki}$ including uniform distribution $U(\mu_{ki}, 1/30)$ on a disc with center at $\mu_{ki}$ and radius 1/30, Gaussian distribution $N(\mu_{ki}, 0.03\mathbf{I})$, and mixed these two distributions, $k = 1, 2$.

## D EXPERIMENTAL SETTING

### D.1 HYPERPARAMETER

We burn in the first 1000 particles for ULA and MALA in all experiments. We initialize the particles in SVGD, ULA and MALA as zeros or random samples from Gaussians. We provide the step size settings in Table 3. For SVGD, we use RBF kernel $k(x, x') = \exp(-\frac{1}{h}\|x - x'\|_2^2)$ and set the bandwidth as $h = \mathrm{med}^2/\log n$, where med is the median of pairwise distances between the particles $\{x_i\}_{i=1}^n$. We set the learning rate of neural networks as 5e-4 for density ratio estimation in REGS. The network structures are presented in Table 4. The initial particles in REGS are sampled from Gaussian distributions.

Table 3: Step size settings for REGS, SVGD, ULA and MALA. "*BGIR*" denotes four datasets including *Banana*, *German*, *Image* and *Ringnorm*.

| Methods | 2Gaussians | 8Gaussians | 9Gaussians | 25Gaussians | BGIR | Covertype |
|---------|-----------|-----------|-----------|------------|------|-----------|
| REGS | 5e-4 | 5e-4 | 5e-4 | 5e-4 | 2e-3 | 2e-3 |
| SVGD | 2e-2 | 2e-2 | 2e-2 | 2e-2 | 5e-2 | 5e-2 |
| ULA | 2e-2 | 5e-2 | 1e-1 | 5e-2 | 1e-3 | 1e-4 |
| MALA | 5e-2 | 2e-1 | 5e-1 | 5e-1 | 1e-3 | – |

### D.2 NEURAL NETWORK ARCHITECTURE

Table 4: Neural network architecture for log-density ratio estimation: feedforward neural networks with equal-width hidden layers and Leaky ReLU activation. Depth $\ell = 3$ for *2Gaussians_1d1*, *2Gaussians_1d2*, *2Gaussians_1d3*, and *2Gaussians*. $\ell = 4$ for *8Gaussians*, *9Gaussians*, *1circle*, *2circles*, *1spiral*, *2spirals*, and *moons*. $\ell = 6$ for *16Gaussians_1c*, *16Gaussians_2c*, *25Gaussians*, *49Gaussians*, and *81Gaussians*.

| Layer | Details | Output size |
|-------|---------|-------------|
| $\{i\}_{i=1}^{\ell-1}$ | Linear, LeakyReLU (0.2) | 128 |
| $\ell$ | Linear | 1 |

### D.3 DATASETS

*Covertype*: https://www.csie.ntu.edu.tw/~cjlin/libsvmtools/datasets/binary.html *Banana, German, Image, Ringnorm*: http://theoval.cmp.uea.ac.uk/matlab/default.html

## E ADDITIONAL NUMERICAL RESULTS

Table 5: Monte Carlo estimates of $\mathbb{E}[h(X)]$. Here $h(x) = \alpha^{\mathrm{T}}x$, $(\alpha^{\mathrm{T}}x)^2$, and $10\cos(\alpha^{\mathrm{T}}x + 1/2)$ with $\alpha \in \mathbb{R}^2$, $\|\alpha\|_2 = 1$. "true" denotes the Monte Carlo estimate with target samples.

| Distributions | $h(x) = \alpha^{\mathrm{T}}x$ | | $h(x) = (\alpha^{\mathrm{T}}x)^2$ | | $h(x) = 10\cos(\alpha^{\mathrm{T}}x + 1/2)$ | |
|---|---|---|---|---|---|---|
| | true | REGS | true | REGS | true | REGS |
| *2Gaussians_1d1* | -0.9886 | -0.9887 | 3.7105 | 3.7093 | 0.46838 | 0.5418 |
| *2Gaussians_1d2* | -0.9972 | -1.1630 | 9.0032 | 9.0297 | -8.3053 | -8.2956 |
| *2Gaussians_1d3* | -0.9602 | -0.728 | 9.8727 | 9.8016 | -6.3785 | -6.1266 |
| *2Gaussians* | -0.0019 | 0.0448 | 8.0243 | 8.0225 | -8.2351 | -8.2513 |
| *8Gaussians* | -0.0021 | -0.021 | 8.0118 | 8.0276 | -3.3925 | -3.3727 |
| *9Gaussians* | 0.0003 | 0.0100 | 10.6771 | 10.8063 | 0.7773 | 0.7532 |
| *16Gaussians_1c* | -0.0008 | -0.0077 | 8.0269 | 8.02889 | -3.4393 | -3.4325 |
| *16Gaussians_2c* | -0.0006 | -0.0060 | 5.3544 | 5.2780 | -1.4260 | -1.3877 |
| *25Gaussians* | 0.0013 | -0.0141 | 8.0276 | 8.0227 | 0.1248 | 0.1189 |
| *49Gaussians* | 0.0006 | 0.0001 | 9.0285 | 9.0215 | 0.2043 | 0.1990 |
| *81Gaussians* | -0.0011 | -0.0009 | 15.0280 | 15.0261 | 0.4126 | 0.4196 |
| *8Gaussians* r=5 | -0.0011 | -0.0037 | 12.5291 | 12.5356 | -1.2160 | -1.2214 |
| *8Gaussians* r=10 | 0.0021 | -0.0030 | 50.0278 | 49.4783 | 3.3872 | 3.4238 |
| *8Gaussians* r=15 | 0.0003 | -0.0306 | 112.5023 | 111.4456 | -1.1116 | -1.0930 |
| *2Gaussians* $\sigma^2 = 0.01$ | -0.0022 | -0.0009 | 0.5119 | 0.5082 | 6.6390 | 6.6504 |
| *2Gaussians* $\sigma^2 = 0.005$ | -0.0012 | 0.0027 | 0.5019 | 0.4976 | 6.6694 | 6.6718 |
| *2Gaussians* $\sigma^2 = 0.0001$ | 0.0005 | 0.0034 | 0.5043 | 0.5049 | 6.6558 | 6.6406 |

Table 6: Monte Carlo estimates of $\mathbb{E}[h(X)]$ by four samplers for 2D mixtures of Gaussians of $X$ with equal weights. Here $h(x) = \alpha^{\mathrm{T}}x$, $(\alpha^{\mathrm{T}}x)^2$ or $10\cos(\alpha^{\mathrm{T}}x + 1/2)$ with $\alpha \in \mathbb{R}^2$, $\|\alpha\|_2 = 1$. "true" denotes the Monte Carlo estimate with target samples. "ULA_$k$" and "MALA_$k$" denote the ULA and MALA with $k$ chains, respectively.

| Distributions | $\sigma^2$ | $h(x) = \alpha^{\mathrm{T}}x$ | | | | | | | $h(x) = (\alpha^{\mathrm{T}}x)^2$ | | | | | | | $h(x) = 10\cos(\alpha^{\mathrm{T}}x + 1/2)$ | | | | | | |
|---|---|---|---|---|---|---|---|---|---|---|---|---|---|---|---|---|---|---|---|---|---|---|
| | | true | REGS | SVGD | ULA_1 | MALA_1 | ULA_50 | MALA_50 | true | REGS | SVGD | ULA_1 | MALA_1 | ULA_50 | MALA_50 | true | REGS | SVGD | ULA_1 | MALA_1 | ULA_50 | MALA_50 |
| 2gaussian | 0.2 | 0.02 | **0.00** | -0.01 | -1.45 | 0.66 | 0.11 | -0.34 | 2.56 | **2.50** | 2.50 | 2.62 | 2.27 | 8.24 | 8.25 | 0.97 | **1.06** | 1.09 | 4.54 | -0.43 | -7.64 | -7.30 |
| | 0.1 | -0.00 | **-0.03** | 0.04 | 1.37 | -1.38 | 0.11 | -0.23 | 2.20 | **2.20** | 2.20 | 2.06 | 2.12 | 8.10 | 8.15 | 1.23 | **1.33** | 1.12 | -2.62 | 5.71 | -7.99 | -7.71 |
| | 0.05 | 0.00 | **0.02** | 0.06 | 1.42 | -1.44 | 0.11 | -0.23 | 2.10 | **2.10** | 2.12 | 2.10 | 2.19 | 8.04 | 8.10 | 1.30 | **1.23** | 1.05 | -3.22 | 5.57 | -8.20 | -7.83 |
| | 0.03 | -0.01 | **0.02** | 0.10 | -1.41 | 1.42 | 0.11 | -0.23 | 2.02 | 2.05 | **2.01** | 2.04 | 2.10 | 8.03 | 8.18 | 1.42 | **1.28** | 1.12 | 5.98 | -3.36 | -8.29 | -7.53 |
| 8gaussian | 0.2 | 0.00 | **-0.01** | 0.00 | -3.46 | 3.00 | 0.14 | 0.35 | 8.20 | **8.20** | 7.98 | 12.81 | 10.20 | 7.87 | 8.05 | -3.05 | **-3.09** | 1.41 | -7.20 | -5.41 | -2.95 | -2.71 |
| | 0.1 | -0.01 | **-0.02** | 0.02 | 2.83 | 2.84 | -0.66 | -0.02 | 8.11 | 8.12 | 8.12 | **8.11** | 8.23 | 8.09 | 8.52 | -3.23 | **-3.26** | 1.51 | -9.35 | -9.11 | -3.54 | -3.71 |
| | 0.05 | -0.01 | **-0.00** | 0.02 | 2.83 | -2.83 | -0.41 | 0.04 | 8.06 | **8.06** | 8.31 | 8.05 | 8.08 | 8.21 | 9.02 | -3.33 | **-3.34** | 1.36 | -9.58 | -6.51 | -2.83 | -4.37 |
| | 0.03 | 0.00 | **-0.00** | -0.01 | -0.00 | 1.96 | -0.41 | 0.18 | 8.04 | **8.05** | 8.05 | 0.02 | 6.19 | 8.19 | 8.40 | -3.35 | **-3.35** | 1.57 | 8.68 | -2.38 | -2.86 | -3.86 |
| 25gaussian | 0.2 | -0.00 | **0.00** | -0.44 | 0.22 | -0.45 | 0.02 | 0.02 | 8.18 | **8.20** | 9.46 | 8.02 | 7.96 | 8.31 | 8.10 | 0.10 | **0.11** | 0.75 | 0.53 | 0.15 | 0.05 | 0.13 |
| | 0.1 | 0.00 | **0.00** | 0.04 | 0.15 | 0.59 | -0.05 | -0.05 | 8.11 | **8.09** | 2.11 | 7.42 | 8.09 | 8.19 | 7.97 | 0.10 | **0.11** | 3.44 | 0.99 | -0.15 | 0.12 | 0.12 |
| | 0.05 | -0.00 | -0.01 | **-0.00** | 0.36 | -1.50 | -0.33 | -0.15 | 8.06 | 7.62 | 1.07 | 4.40 | 6.17 | 5.48 | **8.04** | 0.12 | **0.10** | 5.37 | 0.95 | 0.71 | 0.30 | 0.34 |
| | 0.03 | -0.00 | -0.02 | **-0.01** | 0.14 | -0.08 | 0.14 | -0.04 | 8.04 | 7.58 | 0.98 | **8.11** | 6.97 | 2.95 | 7.41 | 0.11 | **0.09** | 5.56 | -2.40 | -0.91 | 1.62 | 0.08 |

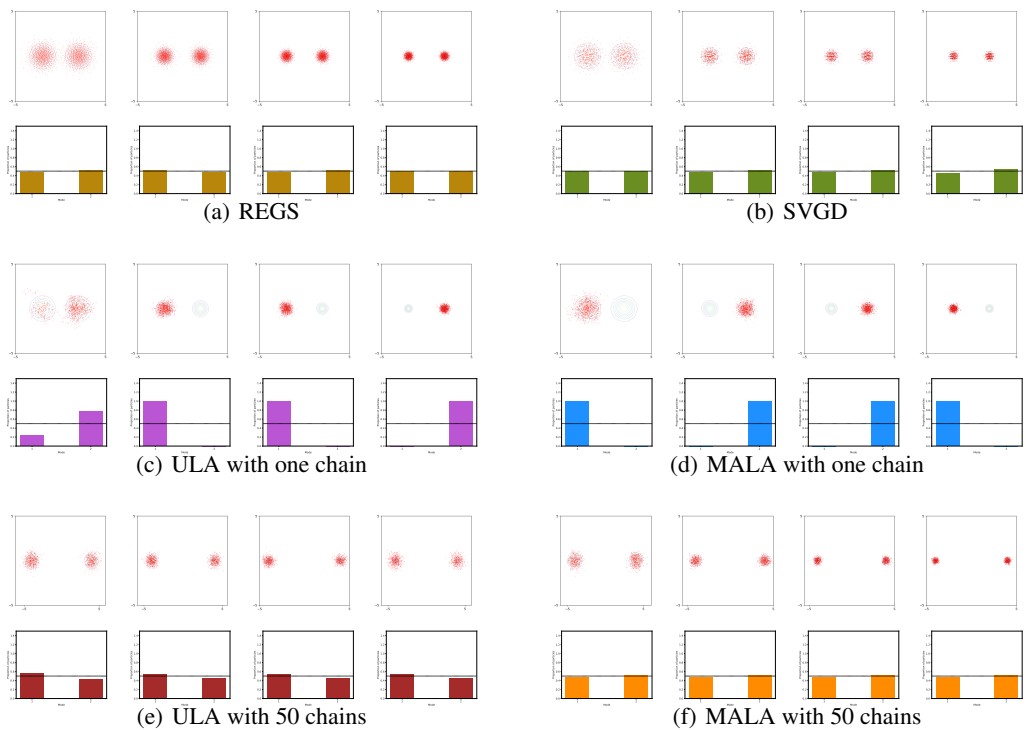

Figure 5: Scatter plots with contours of 1000 generated samples from unnormalized mixtures of 2 Gaussians with equal weights by (a) REGS, (b) SVGD, (c) ULA and (d) MALA with one chain, (e) ULA and (f) MALA with 50 chains. From left to right in each subfigure, the variance of Gaussians varies from $\sigma^2 = 0.2$ (first column), $\sigma^2 = 0.1$ (second column), $\sigma^2 = 0.05$ (third column), to $\sigma^2 = 0.03$ (fourth column).

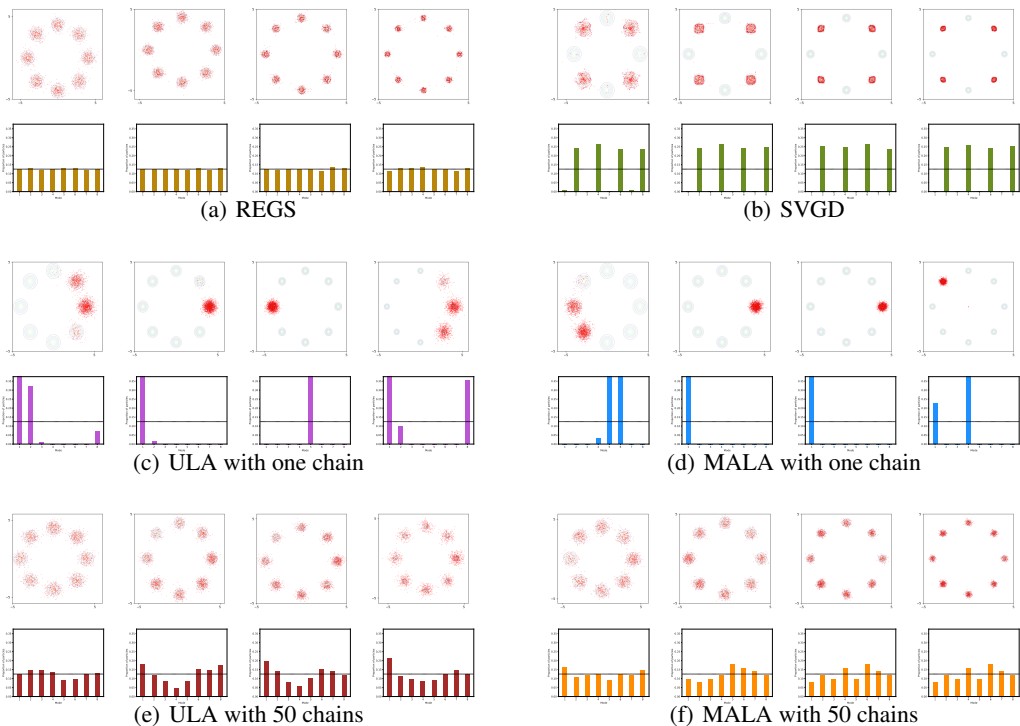

Figure 6: Scatter plots with contours of 2000 generated samples from unnormalized mixtures of 8 Gaussians with equal weights by (a) REGS, (b) SVGD, (c) ULA and (d) MALA with one chain, (e) ULA and (f) MALA with 50 chains. From left to right in each subfigure, the variance of Gaussians varies from $\sigma^2 = 0.2$ (first column), $\sigma^2 = 0.1$ (second column), $\sigma^2 = 0.05$ (third column), to $\sigma^2 = 0.03$ (fourth column).

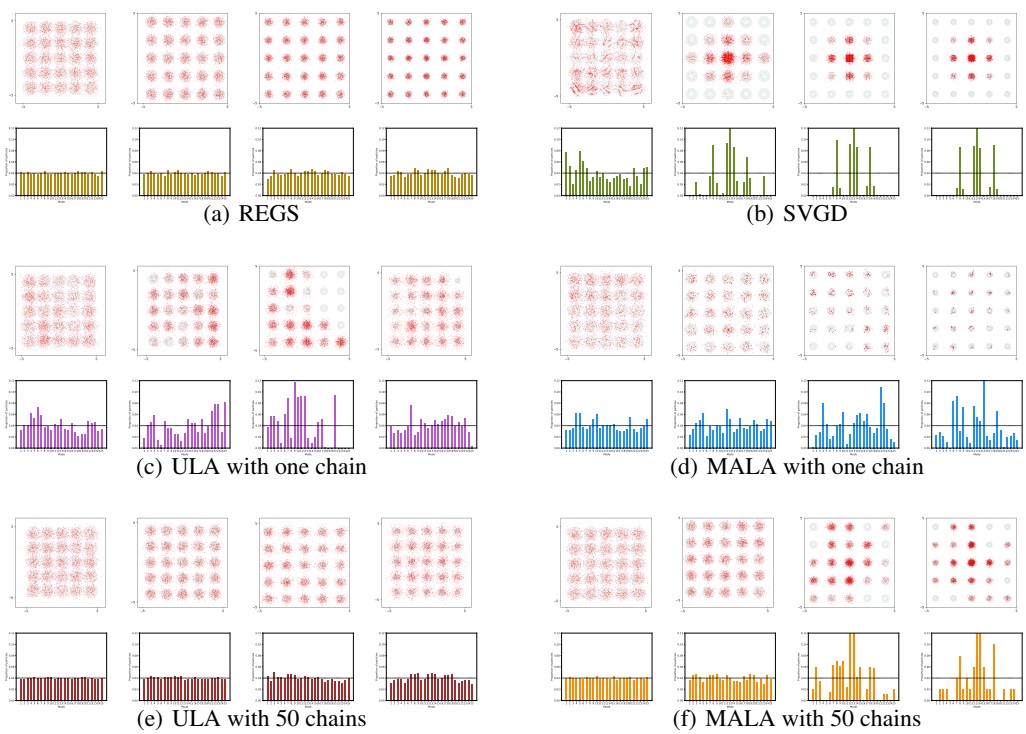

Figure 7: Scatter plots with contours of 5000 generated samples from unnormalized mixtures of 25 Gaussians with equal weights by (a) REGS, (b) SVGD, (c) ULA and (d) MALA with one chain, (e) ULA and (f) MALA with 50 chains. From left to right in each subfigure, the variance of Gaussians varies from $\sigma^2 = 0.2$ (first column), $\sigma^2 = 0.1$ (second column), $\sigma^2 = 0.05$ (third column), to $\sigma^2 = 0.03$ (fourth column).

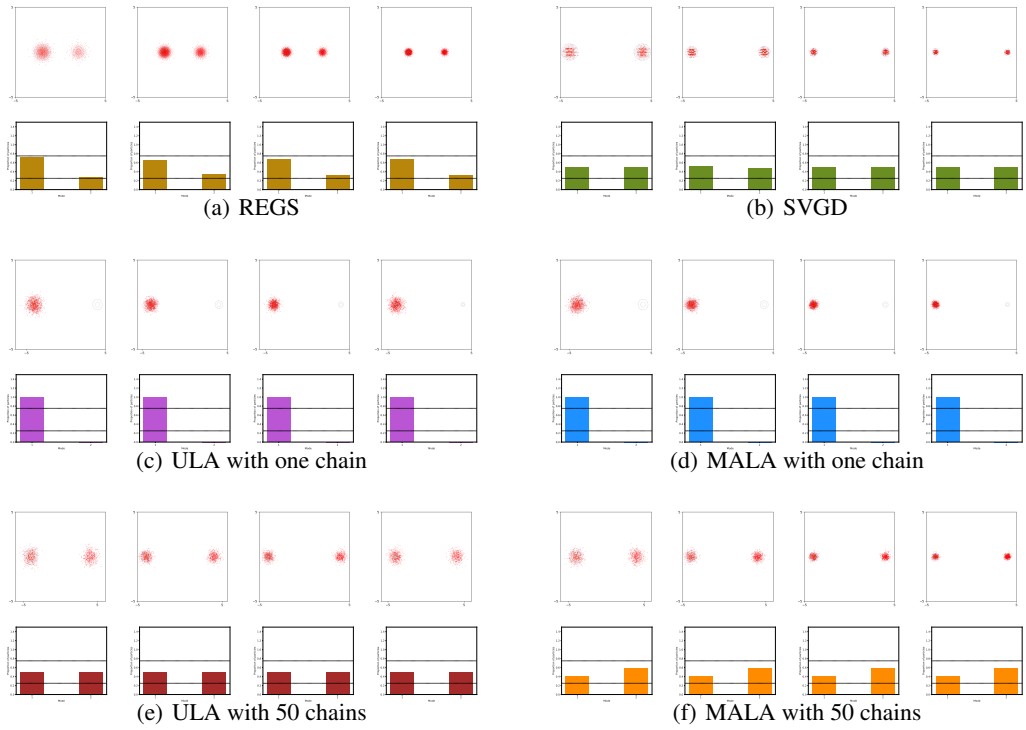

Figure 8: Scatter plots with contours of 1000 generated samples from unnormalized mixtures of 2 Gaussians with unequal weights (0.75, 0.25) by (a) REGS, (b) SVGD, (c) ULA and (d) MALA with one chain, (e) ULA and (f) MALA with 50 chains. From left to right in each subfigure, the variance of Gaussians varies from $\sigma^2 = 0.2$ (first column), $\sigma^2 = 0.1$ (second column), $\sigma^2 = 0.05$ (third column), to $\sigma^2 = 0.03$ (fourth column).

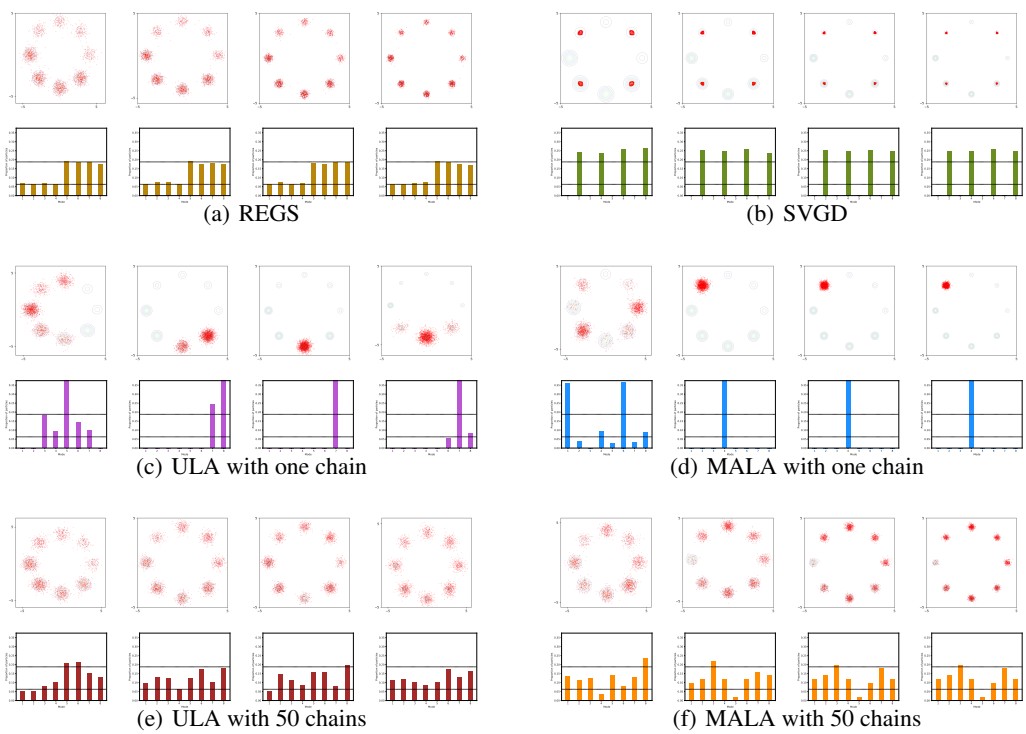

Figure 9: Scatter plots with contours of 2000 generated samples from unnormalized mixtures of 8 Gaussians with unequal weights $(1, 1, 1, 1, 3, 3, 3, 3)/16$ by (a) REGS, (b) SVGD, (c) ULA and (d) MALA with one chain, (e) ULA and (f) MALA with 50 chains. From left to right in each subfigure, the variance of Gaussians varies from $\sigma^2 = 0.2$ (first column), $\sigma^2 = 0.1$ (second column), $\sigma^2 = 0.05$ (third column), to $\sigma^2 = 0.03$ (fourth column).

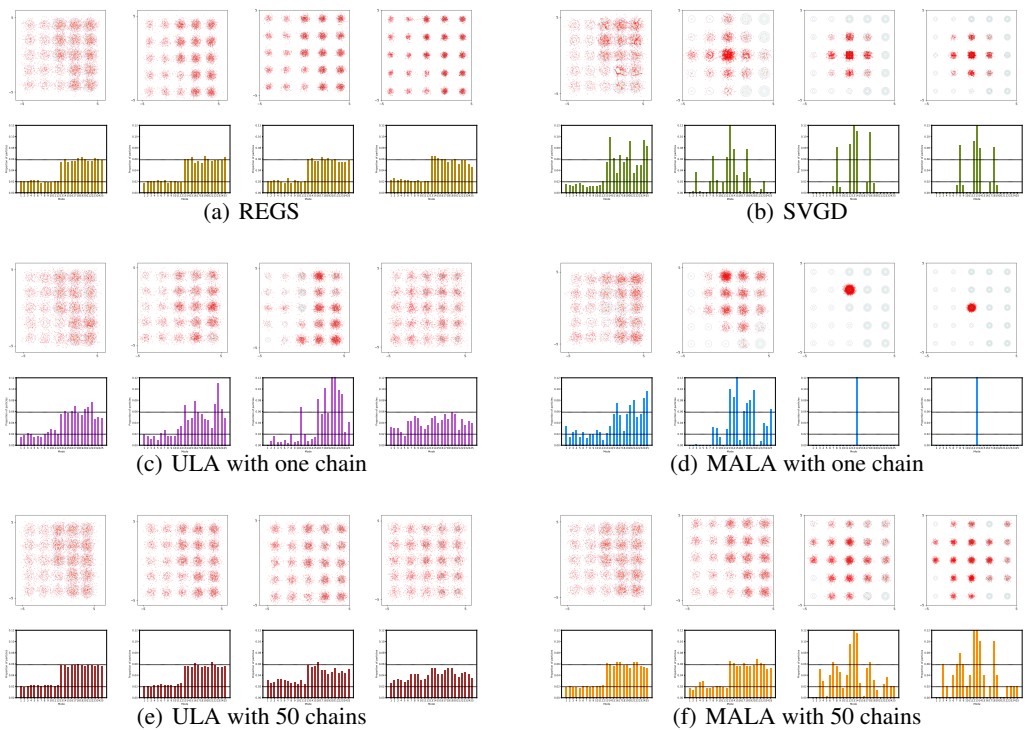

Figure 10: Scatter plots with contours of 5000 generated samples from unnormalized mixtures of 25 Gaussians with unequal weight by (a) REGS, (b) SVGD, (c) ULA and (d) MALA with one chain, (e) ULA and (f) MALA with 50 chains, where each of the first 12 components has weight $1/51$, and each of the rest has weight $3/51$. From left to right in each subfigure, the variance of Gaussians varies from $\sigma^2 = 0.2$ (first column), $\sigma^2 = 0.1$ (second column), $\sigma^2 = 0.05$ (third column), to $\sigma^2 = 0.03$ (fourth column).

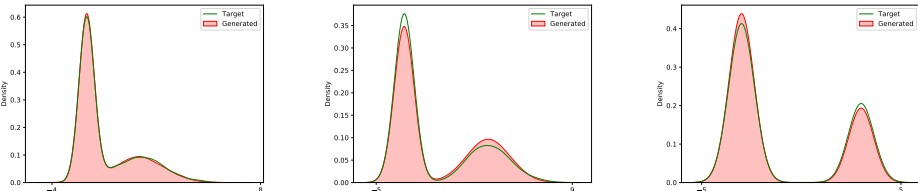

Figure 11: KDE plots for 1D mixtures of 2 Gaussians. Green lines stand for target samples and pink areas represent generated samples by REGS. From left to right, the means and variances of the components changed and the unnormalized densities are given in Scenarios 1, 2, 3 in Appendix B.

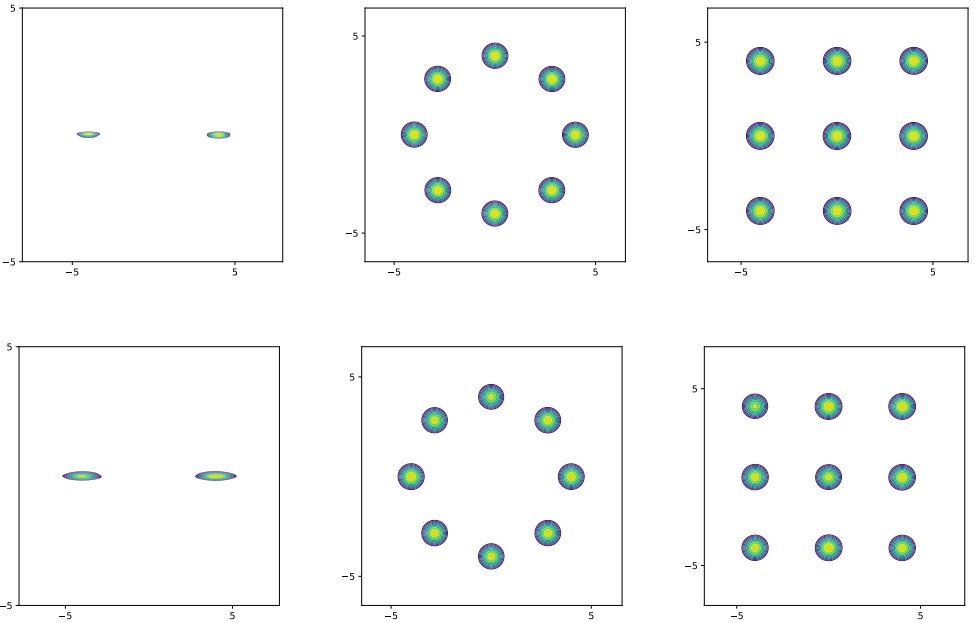

Figure 12: KDE plots of target samples (first row) and generated samples (second row) for two-dimensional mixtures of Gaussians with variance $0.03$. The target samples are from unnormalized density functions $u(x)$ of mixtures of 2 Gaussians in Scenario 4, 8 Gaussians in Scenario 5 and 9 Gaussians in Scenario 6.

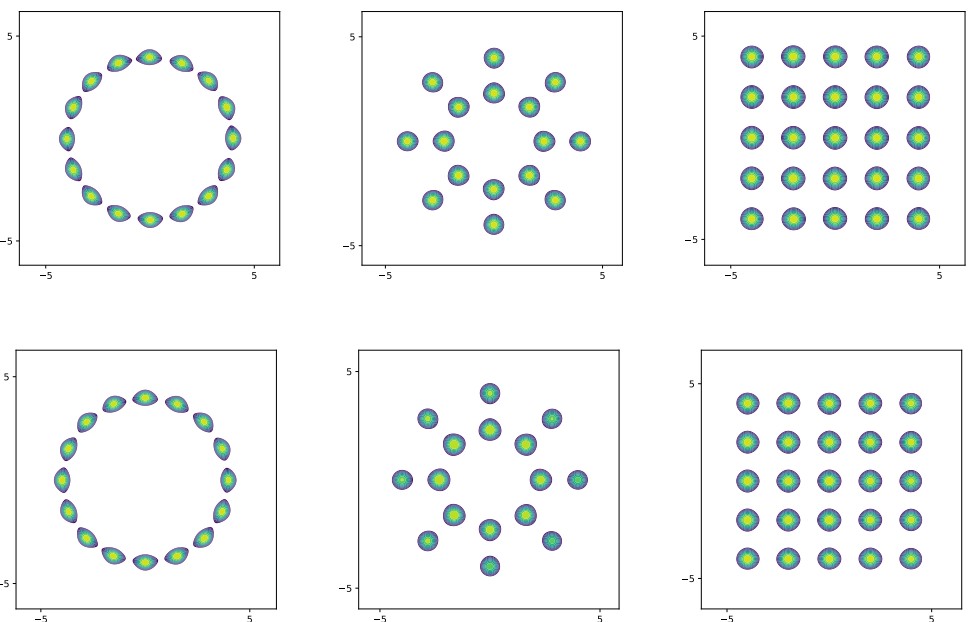

Figure 13: KDE plots of target samples (first row) and generated samples (second row) for 2D mixtures of Gaussians with component variance $0.03$. The corresponding unnormalized densities are presented in Scenarios 7, 8, 9 in Appendix B.

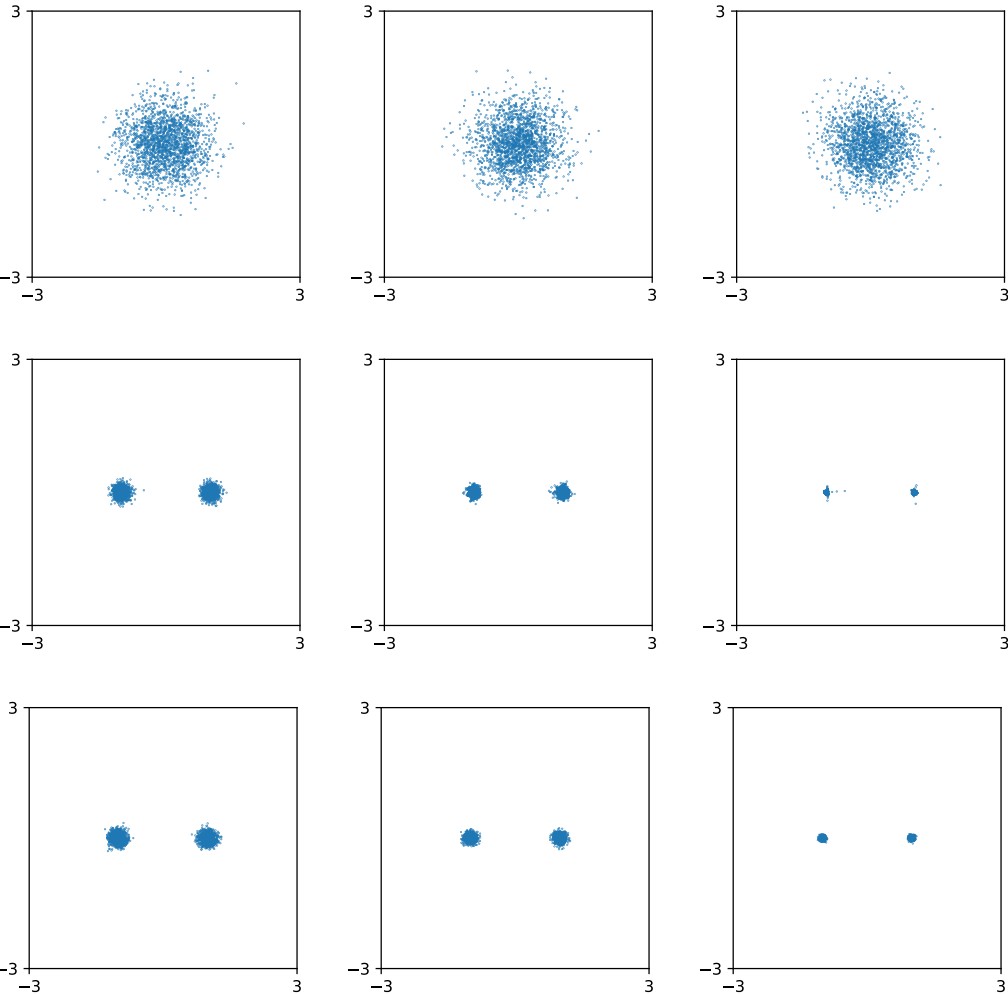

Figure 14: Scatter plots of initial samples (first row), generated samples (second row) for two-dimensional mixtures of 2 Gaussians, and scatter plots of target samples (last row). The target samples are from unnormalized density functions $u(x)$ of mixtures of 2 Gaussians with variance $\sigma^2 = 0.01$ (left column), $\sigma^2 = 0.005$ (middle column) and $\sigma^2 = 0.001$ (right column).

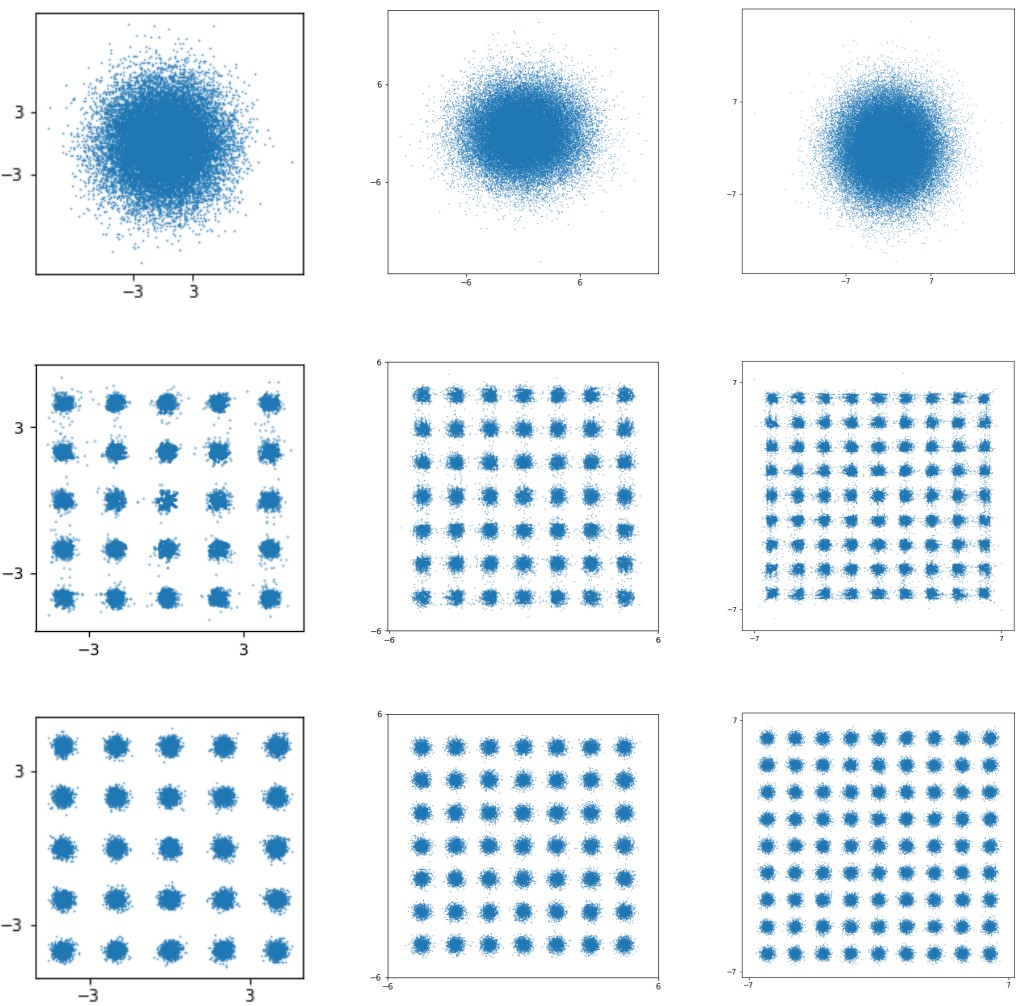

Figure 15: Scatter plots of initial samples (first row), generated samples (second row) for two-dimensional mixtures of multiple Gaussians with variance $\sigma^2 = 0.03$, and scatter plots of target samples (last row). The target samples are from unnormalized density functions $u(x)$ of mixtures of 25 Gaussians (left column), 49 Gaussians (middle column) and 81 Gaussians (right column).

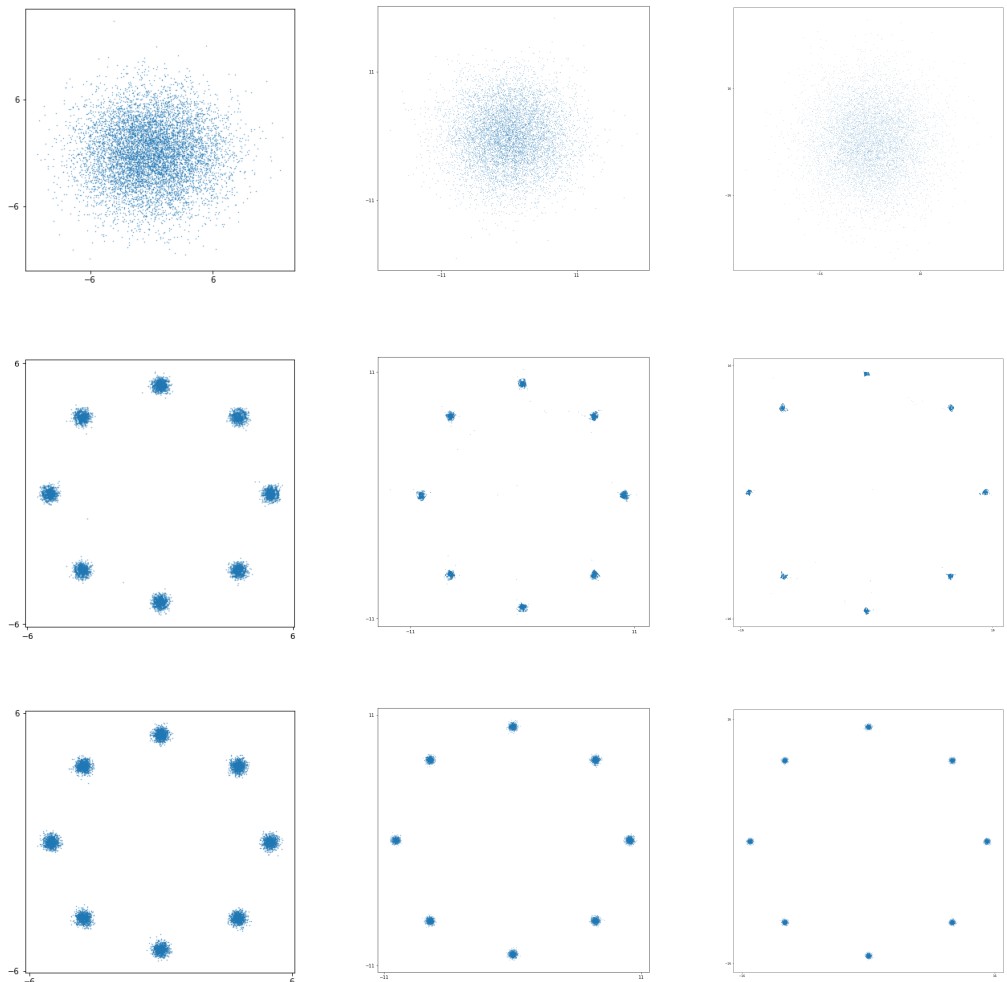

Figure 16: Scatter plots of initial samples (first row), generated samples (second row) for two-dimensional mixtures of 8 Gaussians with variance $\sigma^2 = 0.03$ and varying radius, and scatter plots of target samples (last row). The target samples are from unnormalized density functions $u(x)$ of mixtures of Gaussians with radius being 5 (left column), 10 (middle column) and 15 (right column).

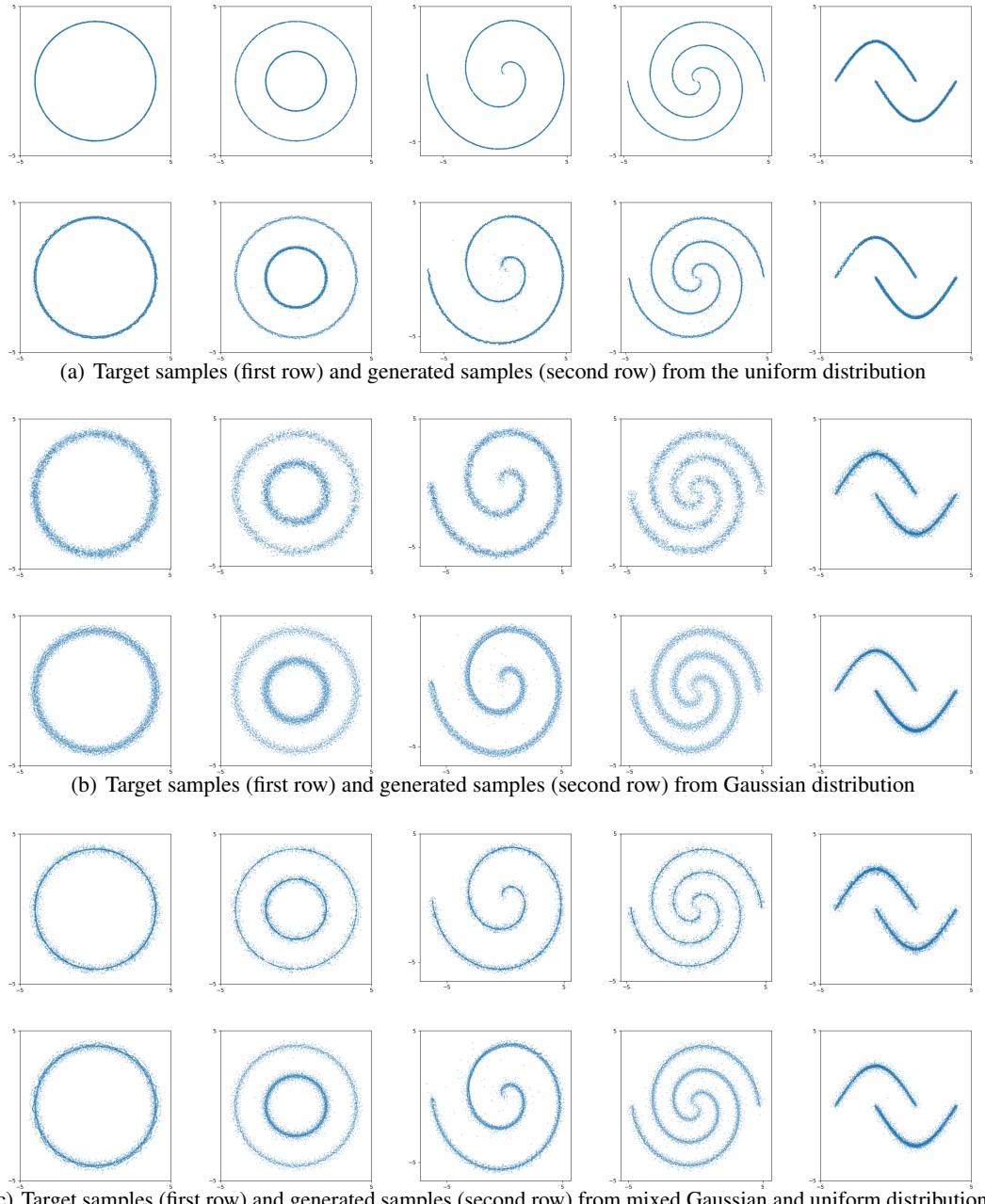

(a) Target samples (first row) and generated samples (second row) from the uniform distribution

(b) Target samples (first row) and generated samples (second row) from Gaussian distribution

(c) Target samples (first row) and generated samples (second row) from mixed Gaussian and uniform distributions

Figure 17: Scatter plots from left to right are one circle (*1circle*, Scenario 12), two circles (*2circles*, Scenario 13), one spiral (*1spiral*, Scenario 14), two spirals (*2spirals*, Scenario 15), and moons (*moons*, Scenario 16).

