# OpenReview forum: "Relative Entropy Gradient Sampler for Unnormalized Distributions"
_ICLR.cc/2022/Conference — ICLR 2022 Submitted_

### Official Review · Reviewer_r2Jt · 2021-10-24

**Correctness:** 3
**Technical Novelty And Significance:** 3
**Empirical Novelty And Significance:** 2
**Recommendation:** 3
**Confidence:** 4

**Main Review:**

Overall, the paper is well written: contributions are clearly stated, relation to previous is presented, the proposed method is well explained and a somewhat extended comparative numerical evaluation of the proposed method is given.

I find the overall approach interesting - the difficulty of sampling being cast as a density ratio estimation problem. This new problem, however, is not easy to solve and your approach of using a neural network to estimate the density ratio seems to work, at least in the considered examples.

That being said, I do have some issues with the paper. You mention in the conclusion that you hope to establish the convergence properties of the proposed method. I understand that it is not trivial to establish convergence. However, not presenting at least some intuition about the convergence of the algorithm is a strong drawback. The way I see it is that there are two sources of error that could hamper convergence: the discretization error of the numerical implementation of the continuous gradient flow and the approximation error of the density ratio. I didn't find anything about either in the paper. If not a proof, at least some intuition about how they affect the results, about how they interact, etc.

In the numerical experiments section, you present a fair amount of examples which show that the proposed method is capable of outperforming the competing algorithms. The results are interesting, however, there are no results to show how the performances of the proposed method vary with the different parameters, notably the number of considered particles, and the choice of distribution w. Such results would cast some light into the inner workings of the proposed method and would be useful for anyone interested in using it.

In section 6.4 you do mention that the improved performances of the proposed method come with a higher computational cost. However, you do not perform any analysis of the trade-off between computation time and performances. It would have also been interesting to compare the performances of the different algorithms for the same computational budget.

Another aspect that struck out to me is the choice of competing algorithms. The algorithms that you choose as competitors are valid, however their choice is questionable. My first remark was why didn't you choose the SMC algorithm as a competitor? It also uses particles in order to estimate the unnormalized target distribution. Also, HMC could have also been considered.

I spotted some typos here and there, for example chians instead of chains on page 8 in the bottom paragraph "... denote the ULA and MALA with k chians", repeats instead of repeat on page 9 "We repeats the random partition 10 times.".

Another small issue is with figure 4, it's hardly readable. I understand that there is a limit on the page count, however, that's not a justification for having figures that are hard to read. More so, as there are some redundancies in the text that could have been eliminated, for ex. equations (11) and (13) are the same, is the presence of both necessary for understanding the idea that is presented in section 4?

**Summary Of The Paper:**

The paper addresses the issue of sampling from an unnormalized distribution. The sampling problem is cast as the numerical simulation of the gradient flow associated with the KL divergence between the target unnormalized distribution and the approximating distribution. The challenging part is to estimate the density ratio that appears in the gradient term. The authors propose to use a deep neural network to estimate the density ratio. Numerical results show the usefulness of the proposed method.

**Summary Of The Review:**

I find the approach interesting. However, there are some issue with the paper as it is, both theoretical and empirical. From a theoretical point of view, there is no discussion about conditions for convergence of the algorithm. From an empirical point of view, the numerical experiments are not complete enough, with respect to the comparison analysis that is carried out, but also with respect to compensating missing theoretical analysis. Overall, the paper is interesting, but in its current form is not ripe enough for publication.

---

> ### Author Response · Authors · 2021-11-22
> **Response to Reviewer r2Jt**
>
> Thank you for taking the time to review our paper.
>
> "Overall, the paper is well written: contributions are clearly stated, relation to previous is presented, the proposed method is well explained and a somewhat extended comparative numerical evaluation of the proposed method is given.
>
> I find the overall approach interesting - the difficulty of sampling being cast as a density ratio estimation problem. This new problem, however, is not easy to solve and your approach of using a neural network to estimate the density ratio seems to work, at least in the considered examples.''
>
> Thank you so much for noticing the merits of our work.
>
> "That being said, I do have some issues with the paper. You mention in the conclusion that you hope to establish the convergence properties of the proposed method. I understand that it is not trivial to establish convergence. However, not presenting at least some intuition about the convergence of the algorithm is a strong drawback. The way I see it is that there are two sources of error that could hamper convergence: the discretization error of the numerical implementation of the continuous gradient flow and the approximation error of the density ratio. I didn't find anything about either in the paper. If not a proof, at least some intuition about how they affect the results, about how they interact, etc."
>
> We will add the theoretical analysis part in an updated version.
>
> "In the numerical experiments section, you present a fair amount of examples which show that the proposed method is capable of outperforming the competing algorithms. The results are interesting, however, there are no results to show how the performances of the proposed method vary with the different parameters, notably the number of considered particles, and the choice of distribution w. Such results would cast some light into the inner workings of the proposed method and would be useful for anyone interested in using it."
>
> Thank you for your insightful comments. We will show the performances of our method is consistent across different parameters. In our implementation, the number of particles and the choice of the reference distribution needs to be tuned for target distributions with different dimensions of support spaces. For the numerical parameters such as step size, we use the same value for different models considered in the paper,  including the mixtures of Gaussians, multivariate Gaussians, and the Bayesian logistic regression. For the parameters for training neural networks like the learning rate, we select them following the common practice in neural network computing. For example, the learning rate should not be too large. We will provide some discussions about parameter selection in an updated version of the paper.
>
>
> "In section 6.4 you do mention that the improved performances of the proposed method come with a higher computational cost. However, you do not perform any analysis of the trade-off between computation time and performances. It would have also been interesting to compare the performances of the different algorithms for the same computational budget."
>
> We will include the comparison of running time of the considered algorithms. In some Gaussian mixture examples, the performance of ULA and MALA is affected by the metastability issue. Therefore, ULA and MALA will not perform better by letting them run for longer time. We will add more experimental results to demonstrate this.
>
> "Another aspect that struck out to me is the choice of competing algorithms. The algorithms that you choose as competitors are valid, however their choice is questionable. My first remark was why didn't you choose the SMC algorithm as a competitor? It also uses particles in order to estimate the unnormalized target distribution. Also, HMC could have also been considered."
>
> We believe the SMC algorithm and HMC algorithm are less related to our method since these algorithms are not directly based gradient flows in spaces of probability measures. However, we will be happy to include these two algorithms in the experimental comparison in the revision.
>
> "I spotted some typos here and there, for example chains instead of chains on page 8 in the bottom paragraph "... denote the ULA and MALA with k chains", repeats instead of repeat on page 9 "We repeats the random partition 10 times."."
>
> Thank you for pointing out the typos and we will correct them.
>
> "Another small issue is with figure 4, it's hardly readable. I understand that there is a limit on the page count, however, that's not a justification for having figures that are hard to read. More so, as there are some redundancies in the text that could have been eliminated, for ex. equations (11) and (13) are the same, is the presence of both necessary for understanding the idea that is presented in section 4?"
>
> We will update the figure 4 to make it more readable. Meantime, we will condense equations (11) and (13) into a single equation.

---

> > ### Comment · Reviewer_r2Jt · 2021-11-30
> > **Read the author rebuttal**
> >
> > I read the author's response.
> >
> > Unfortunately, I still have to stick with my initial assessment of rejecting the paper. The issue that I have is that after all the modifications the authors said they will carry, the paper will be too different from the originally submitted one. Thus, I am required to review a paper I have yet to read.
> > I feel sorry for the authors, however, as I said in my review, the paper as it was submitted is not ripe enough for publication.

---

### Official Review · Reviewer_opqD · 2021-11-02

**Correctness:** 1
**Technical Novelty And Significance:** 2
**Empirical Novelty And Significance:** 4
**Recommendation:** 5
**Confidence:** 3

**Main Review:**

I enjoyed reading the paper. It is well written, the motivation is clear and it is easy to follow the main ideas.

However, I find it hard to assess the actual contribution of the paper.
On one hand, while the proposed algorithm makes sense, there is no guarantee in the paper, either for the sampling accuracy or even for the fact that the algorithm will converge to the target measure. For example, is there any guarantee that the discretized flow does not add bias to the obtained measure?
There are many tunable parameters in the algorithms, $s$ the discretization step, $K$, the time horizon, $n$, the number of particles. What is the interplay between those parameters, given some distribution, how should I choose those?
I would have expected to see a bit more of the underlying theory behind this algorithm.

On the other hand, from the perspective of actual results, I find that the numerical experiments are somewhat restricted and artificial. Coupled with the computational overhead, it's not clear to me when one will actually prefer to use the new algorith,


**Summary Of The Paper:**

The paper proposes a novel way to sample from unnormalized distributions. This is helpful when calculating or estimating the normalizing constant is untractable.

The main idea is to track the gradient flow of the relative entropy in the Wasserstein space of probability distributions. It is known that the flow converges to the target distribution and the paper introduces a variational characterization of the discretized steps. The main benefit of this characterization is that it bypasses the need to know the normalizing constant as well as being amenable to estimation by using a combined particle evolution.

The benefits of the new algorithm are demonstrated through several numerical simulations.

**Summary Of The Review:**

The idea is elegant interesting but the paper lacks evidence for its usefulness, both from theoretical and applied perspectives.

---

> ### Author Response · Authors · 2021-11-22
> **Response to Reviewer opqD**
>
> We appreciate your taking the time to review our paper.  We thank you for your comments that our paper is well written and the idea is novel.
>
> Convergence analysis of the proposed algorithm is highly nontrivial. Nonetheless, we have made some progress since submitting the paper. We will include some convergence analysis results in an updated version of the paper.
>
>  Most of the tuning parameters are commonly used in the gradient flow literature, such as [1] and [2]. We will also include some discussions about the choices of parameters. The experimental results are shown to illustrate our method can overcome the problem of metastability which greatly affects the performance of ULA and MALA. However, we are happy to include more experimental comparison as you suggested.
>
> [1] Qiang Liu and Dilin Wang (2016). Stein Variational Gradient Descent: A General Purpose Bayesian Inference Algorithm, in NIPS.
>
> [2] Youssef Mroueh, Tom Sercu, Anant Raj (2019). Sobolev Descent, in AISTATS.

---

### Official Review · Reviewer_dwfT · 2021-11-03

**Correctness:** 3
**Technical Novelty And Significance:** 2
**Empirical Novelty And Significance:** 2
**Recommendation:** 5
**Confidence:** 4

**Main Review:**

Wasserstein gradient flow has proved to be a useful tool for sampling from an unnormalized distribution. Section 2 to 4 of this work follow the standard derivation of the work along this research line and well explain the particle evolution strategy. Since the underlying velocity field of the Wasserstein gradient flow requires the access to the variable distribution $q_t$ which is in general not available, a key step in methods along this research line is to estimate such a quantity. To estimate such a quantity, this work proposes to estimate the log density ratio between the potential function and the variable distribution $q_t$ by minimizing the Bregman score which is described in Section 5.
However, I find Section 5 difficult to understand. It would be vary helpful if the authors could explain the intuition of the Bregman score. In fact, I think there should be an individual section in the preliminary that describes the Bregman score and all the statements below equation (15) so that the reader can follow this very important step.

I think section 5 is the part that differs this work from previous work like [1] and is where the novelty of this paper lies. It need to be very clearly explained.


[1] Degond, Pierre, and Francisco-José Mustieles. "A deterministic approximation of diffusion equations using particles." SIAM Journal on Scientific and Statistical Computing 11, no. 2 (1990): 293-310.

**Summary Of The Paper:**

This paper considers the problem of sampling from an unnormalized distribution. The unnormalized target distribution can be regarded as a stationary point of the Wasserstein gradient flow of the corresponding relative entropy functional, which can be equivalently identified from a microscopic perspective by defining a time-varying velocity field of the particles. While the exact time-varying velocity field is not exactly available, the authors propose to estimate such a quantity by approximating the corresponding logarithmic density ratio through minimizing the Bregman score. Such an approximation requires only samples from the variable distribution which can obtain by simulating particles following the estimated velocity field.

**Summary Of The Review:**

This paper leverage the microscopic equivalence of the Wasserstein gradient flow of the relative entropy to sample from an unnormalized distribution, but the derivation of the key step in the proposed approach is not well explained.

---

> ### Author Response · Authors · 2021-11-22
> **Response to Reviewer dwfT**
>
> We thank you for taking the time to review our paper.
>
> "However, I find Section 5 difficult to understand. It would be very helpful if the authors could explain the intuition of the Bregman score. In fact, I think there should be an individual section in the preliminary that describes the Bregman score and all the statements below equation (15) so that the reader can follow this very important step.
> I think section 5 is the part that differs this work from previous work like [1] and is where the novelty of this paper lies. It need to be very clearly explained.
>
> [1] Degond, Pierre, and Francisco-José Mustieles. "A deterministic approximation of diffusion equations using particles." SIAM Journal on Scientific and Statistical Computing 11, no. 2 (1990): 293-310."
>
>
> Thank you for your comments. We will revise Section 5 to make it more readable. Particle methods are commonly used to obtain numerical solutions of diffusion equations [1]. However, the study of particle methods in [1] lies in the scope of numerical simulation of a given equation. In our work, the equation is derived from the minimization of KL divergence over the space of probability measures. The objectives and the motivations are very different. Furthermore, the treatments of velocity fields are different. We estimate the velocity fields by the proposed density ratio estimation method. This work [1] used simple smoothed approximation methods with various cutoff functions. We will clarify the differences in our updated version.

---

> > ### Comment · Reviewer_dwfT · 2021-11-29
> > **Response to the Reply**
> >
> > After reading the response from the authors, my concerns are not address and hence my score remains unchanged. I think the authors need to clearly state and verify the advantage of their approach over existing techniques like [1].

---

### Official Review · Reviewer_g4HT · 2021-11-03

**Correctness:** 2
**Technical Novelty And Significance:** 2
**Empirical Novelty And Significance:** 2
**Recommendation:** 1
**Confidence:** 5

**Main Review:**

The paper appears to miss the fact that the 2-Wasserstein gradient flow for the relative entropy defines a Markov process, which is exactly the Langevin dynamics; this can be seen by comparing Eq. (4) to the Fokker–Planck equation for an Ito diffusion (e.g., Eq. (4.1) in [1]; see also section 3.5 in [2] for a relevant discussion from the ML literature). Indeed, the Fokker–Planck equation determines the diffusion uniquely because the differential operator in the Fokker–Planck equation is the adjoint of the infinitesimal generator of the diffusion. Thus, the proposed algorithm is a rather unorthodox way of approximating a Langevin distribution.

The paper makes the said approximation more difficult than it should be by using a particle approximation that requires estimating density ratios, a notorious tricky problem. The algorithm ends up having some ability to handle multimodality because the use of weighted particles and density ratio estimation allows the algorithm to effectively compute the relative volumes of different modes of the distribution. The proposed method for estimating the density ratios appears to be the same as Geyer’s reverse logistic regression method [3], with a neural net replacing the inner product. Thus, I expect similar results could be obtained more directly, and with only a single round of volume estimation, by (1) running MALA many times and (2) then estimating the relative volume of each chain (note there are numerous other methods for doing this other than reverse logistic regression). Such an approach should work well on the kinds of low-dimensional examples considered in the numerical experiments.

Further issues arise in the experimental evaluation. First, the experiments seem to show very similar performance to MALA, all within the standard errors when provided (e.g., in Table 2). Second, I’m concerned about the quality of the MALA implementation, for which code was not included. The lack of convergence in one example suggests MALA was not run with appropriate step size adaptation targeting the optimal acceptance rate [4,5], as is standard in the literature. If so, then the comparison is not appropriate. Third, for a fair comparison, the MALA chains should also be reweighted based on volume estimates for each chain, as described above.

Is it possible there are some gains from using the proposed method on multimodal distribution? Yes. But I remain skeptical. Moreover, if the goal is prediction, I expect combining MCMC with stacking will be more effective [6,7].


[1] Pavliotis, G. A. Stochastic Processes and Applications. (Springer, 2014).

[2] Liu, Q. Stein Variational Gradient Descent as Gradient Flow. In NeurIPS (2017).

[3] Geyer, C. Estimating normalizing constants and reweighting mixtures. Technical Report (1994).

[4] Roberts, G. O. & Rosenthal, J. S. Optimal scaling of discrete approximations to Langevin diffusions. Journal of the Royal Statistical Society: Series B (Statistical Methodology) 60, 255–268 (1998).

[5] Roberts, G. O. & Rosenthal, J. S. Optimal scaling for various Metropolis-Hastings algorithms. Statistical Science 16, 351–367 (2001).

[6] Yao, Y., Vehtari, A. & Gelman, A. Stacking for Non-mixing Bayesian Computations: The Curse and Blessing of Multimodal Posteriors. arXiv.org, arXiv:2006.12335 (2020).

[7] Yao, Y., Vehtari, A., Simpson, D. & Gelman, A. Using Stacking to Average Bayesian Predictive Distributions. Bayesian Analysis 13, 917–1007 (2017).

**Summary Of The Paper:**

The paper suggests a method for approximating the 2-Wasserstein gradient flow for the relative entropy. The proposed particle-based method uses a neural network function approximation-based approach to estimating the necessary density ratios. Experiments verify reasonable performance compared to MALA and ULA.

**Summary Of The Review:**

The paper seems to have a fundamental misunderstanding of the Wasserstein gradient flow for the relative entropy, and the experimental evaluations may not be appropriate.

---

> ### Author Response · Authors · 2021-11-22
> **Response to Reviewer g4HT**
>
> We appreciate your taking the time to read our paper. However, based on your comments, we believe you did not understand our paper at all and are fundamentally ignorant about the basics of Wasserstein gradient flows. Most of your comments are either incorrect or do not faithfully reflect the contents and contributions of our paper. Below we will explain the main idea of our paper and the basic concepts of gradient flows when addressing your specific comments. At the end of our response to your comments below, we also provide some reading materials related to gradient flows and the content of our paper. We encourage you to read them and hope it may help you gain some basic understanding of gradient flows, so that you can perhaps avoid making ignorant comments about other people’s work in the future.
>
> “The paper appears to miss the fact that the 2-Wasserstein gradient flow for the relative entropy defines a Markov process, which is exactly the Langevin dynamics; this can be seen by comparing Eq. (4) to the Fokker–Planck equation for an Ito diffusion (e.g., Eq. (4.1) in [1]; see also section 3.5 in [2] for a relevant discussion from the ML literature). Indeed, the Fokker–Planck equation determines the diffusion uniquely because the differential operator in the Fokker–Planck equation is the adjoint of the infinitesimal generator of the diffusion. Thus, the proposed algorithm is a rather unorthodox way of approximating a Langevin distribution.”
>
> Your comments show a fundamentally lack of understanding of 2-Wassertein gradient flow.
> The 2-Wasserstein gradient flow of the relative entropy has a probabilistic representation with an Ito stochastic differential equation, which is known as the overdamped Langevin diffusion or Langevin dynamics. The 2-Wasserstein gradient flow of the relative entropy can also be formulated as a continuity equation (e.g., equation (4) in our paper). In fluid dynamics, the continuity equation is referred to as the Eulerian coordinates for representing the evolution of probability density functions of particles / samples. Meanwhile, one can use the corresponding Lagrangian coordinates (e.g., equation (7) in our paper) to describe the motion of particles / samples. Connections between the Eulerian description and the Lagrangian description of 2-Wasserstein gradient flows is clearly described in the book by Ambrosio, Gigli, and Savare [2], see, for example, Lemma 8.1.4 on page 172 and Proposition 8.1.8 on page 175 of this book. This connection can be used to handle the computational aspects of 2-Wasserstein gradient flows. In our paper, we use the Lagrangian description to formulate the sampling process from unnormalized distributions, while our algorithm is based on the forward Euler discretization of equation (7) .
>
> Eq 4.1 in [12] provides a general form of the time-homogeneous diffusion process. There are no formal discussions about the Wasserstein gradient flows in this book [12]. Section 3.5 in the paper [13] is about the Stein variational gradient descent (SVGD), which is included in the comparative experiments in our paper. We note that this paper [13]  stated that “Theory of SVGD is parallel to that of Langevin dynamics in many perspectives, but with importance differences”. A brief discussion on the differences was also provided in this paper [13].
>
> “The paper makes the said approximation more difficult than it should be by using a particle approximation that requires estimating density ratios, a notorious tricky problem. The algorithm ends up having some ability to handle multimodality because the use of weights particles and density ratio estimation allows the algorithm to effectively compute the relative volumes of different modes of the distribution. The proposed method for estimating the density ratios appears to be the same as Geyer’s reverse logistic regression method [3], with a neural net replacing the inner product.”
>
> Apparently, you totally misunderstood our work and mistakenly thought that our method is related to Geyer’s reverse logistic regression method [14].  Geyer’s paper proposed the reverse logistic regression method for estimating normalizing constants with samples from unnormalized densities. In contrast, our density ratio estimation method is for estimating the velocity field function, which turns out to be the logarithm of the density ratio between a given unnormalized density function and the density function for particles / samples. These are two completely different problems.
>
> In the density ratio estimation method presented in our paper, we do not need to estimate or compute the so-called relative volumes of different modes of the distribution. Moreover, all the particles are initialized with random samples from a given source distribution (e.g., Gaussian) and get updated using the forward Euler iterations as illustrated in Algorithm 1 in our paper.
>
> What do you mean by “weights particles” in your comments? We did not use this term in our paper.

---

> > ### Author Response · Authors · 2021-11-22
> > **Response to Reviewer g4HT (II)**
> >
> > “Thus, I expect similar results could be obtained more directly, and with only a single round of volume estimation, by (1) running MALA many times and (2) then estimating the relative volume of each chain (note there are numerous other methods for doing this other than reverse logistic regression). Such an approach should work well on the kinds of low-dimensional examples considered in the numerical experiments.”
> >
> > Our method is a nonparametric approach. We do not specify any structure of the target unnormalized density in our approach. In the experiment with Gaussian mixture models, we only use the target density function u to evaluate u(x). We do not use the knowledge of the number of Gaussian components and the relative ratios between the weights of the components.
> >
> > We would appreciate if you could provide more specifics about the sampling method you suggested in your comments. Also, we would be interested in comparing your suggested method with our proposed method in additional experiments if you could make your code publicly available. We will make all our code and detailed description of the experimental setup, including models and tuning parameters, publicly available on GitHub or a platform that you prefer.
> >
> > “Further issues arise in the experimental evaluation. First, the experiments seem to show very similar performance to MALA, all within the standard errors when provided (e.g., in Table 2).”
> >
> > We report the Monte Carlo estimates of the expectation of three different test functions in Table 1 for mixtures of Gaussian distributions. As shown in Table 1, our method performs consistently better than the other methods. According to Figure 4 for multivariate Gaussian distributions, our method performs better as well.
> >
> > Indeed, in the Bayesian logistic regression example, all the methods perform similarly. We believe that the posterior distribution is unimodal and close to be log-concave, hence any reasonable methods would perform well. We noted that ULA and MALA with 50 chains do not show any obvious improvement above ULA and MALA with a single chain.
> >
> > “Second, I’m concerned about the quality of the MALA implementation, for which code was not included. The lack of convergence in one example suggests MALA was not run with appropriate step size adaptation targeting the optimal acceptance rate [4,5], as is standard in the literature. If so, then the comparison is not appropriate. Third, for a fair comparison, the MALA chains should also be reweighted based on volume estimates for each chain, as described above.”
> >
> > We use the source code from [https://github.com/Tom271/LangevinMC] for ULA and MALA. We use the default step size provided in the code, and manually tune the step size parameter for ULA and MALA carefully. We reported the best empirical results of ULA and MALA in our paper.
> >
> >
> > “Is it possible there are some gains from using the proposed method on multimodal distribution? Yes. But I remain skeptical. Moreover, if the goal is prediction, I expect combining MCMC with stacking will be more effective [6,7].”
> >
> > We have provided the demo code for sampling from Gaussian mixtures in the supplemental material. Our method convincingly performs better than ULA and MALA (both use 50 chains). We are happy to compare the mentioned method with our method for the fairness of the experimental evaluations.

---

> > > ### Author Response · Authors · 2021-11-22
> > > **Response to Reviewer g4HT (III)**
> > >
> > > Some recommended reading materials are included below in addition to the cited references in our response:
> > >
> > > [1] Wuchen Li, Stanley Osher (2018). Constrained dynamical optimal transport and its Lagrangian formulation, https://arxiv.org/abs/1807.00937.
> > >
> > > [2] Luigi Ambrosio, Nicola Gigli and Giuseppe Savaré (2008). Gradient flows: in metric spaces and in the space of probability measures. Springer.
> > >
> > > [3] Qiang Liu and Dilin Wang (2016). Stein Variational Gradient Descent: A General Purpose Bayesian Inference Algorithm, in NIPS.
> > >
> > > [4] Youssef Mroueh, Tom Sercu, Anant Raj (2019). Sobolev Descent, in AISTATS.
> > >
> > > [5] Antoine Liutkus, Umut Simsekli, Szymon Majewski, Alain Durmus, Fabian-Robert Stöter (2019). Sliced-Wasserstein Flows: Nonparametric Generative Modeling via Optimal Transport and Diffusions, in ICM.
> > >
> > > [6] Yuan Gao, Yuling Jiao, Yang Wang, Yao Wang, Can Yang, Shunkang Zhang (2019). Deep generative learning via variational gradient flow, in ICML.
> > >
> > > [7] Michael Arbel, Anna Korba, Adil Salim, Arthur Gretton (2019). Maximum Mean Discrepancy Gradient Flow, in NeurIPS 2019.
> > >
> > > [8] Abdul Fatir Ansari, Ming Liang Ang, Harold Soh (2021).  Refining deep generative models via discriminator gradient flow, in ICLR.
> > >
> > > [9]David Alvarez-Melis, Nicolò Fusi (2021). Dataset Dynamics via Gradient Flows in Probability Space, in ICML 2021.
> > >
> > > [10]Anna Korba, Pierre-Cyril Aubin-Frankowski, Szymon Majewski, Pierre Ablin (2021).  Kernel Stein Discrepancy Descent, in ICML 2021.
> > >
> > > [11] Pierre Glaser, Michael Arbel, Arthur Gretton (2021). KALE Flow: A Relaxed KL Gradient Flow for Probabilities with Disjoint Support, in Neurips.
> > >
> > > [12] Grigorios A. Pavliotis (2014). Stochastic Processes and Applications, Springer.
> > >
> > > [13] Qiang Liu (2017). Stein Variational Gradient Descent as Gradient Flow, in NIPS.
> > >
> > > [14] Charles J. Geyer (1994).  Estimating normalizing constants and reweighting mixtures, Technical report, University of Minnesota.

---

> > > > ### Comment · Reviewer_g4HT · 2021-11-22
> > > > **Reply to response**
> > > >
> > > > Thank you for your thorough response. However, you seem to be missing my central concerns and claims:
> > > >
> > > > 1. In your response regarding the 2-Wasserstein gradient flow for the relative entropy, you never refute my central claim: whatever particular representation you may choose to use (in terms of $X_t$ or $q_t$, for example), its unique solution is a Langevin diffusion. How can you then not mention this crucial fact – that you are approximating the Langevin diffusion – in your paper?
> > > >
> > > > 2. My reference to Eq. (4.1) in Pavliotis (2014) was just to the form of the Fokker--Planck equation for the Langevin diffusion; I'm well aware that book is not about gradient flows. My reference to Liu (2017) was just an ML reference to the fact that the solution to the 2-Wasserstein gradient flow for the relative entropy is the Langevin diffusion.
> > > >
> > > > 3. You are right that what you are doing is not quite reverse logistic regression – I should have double checked that. Rather, your approach falls under the framework described in [SSK12] (as does reverse logistic regression). In any case, my main point stands: high-dimensional density (ratio) estimation and normalization constant estimation are difficult problems. If your method requires solving either, then the methods you compare against should compete on a level playing field. Hence, you should compare to using MALA with stacking or something similar.
> > > >
> > > > 4. Thank you for the pointer to the MALA code you used. However, manual tuning of the step size is not appropriate when robust adaptive methods for targeting the optimal acceptance rate are available.
> > > >
> > > > 5. Regarding your method's performance for a multimodal distribution, I'm not claiming it didn't perform well on the simple, 2D synthetic examples from your paper. However, (A) the comparison to a naive MALA implementation (see my #4 above) is not appropriate and (B) this result says almost nothing about how the method would perform on realistic multimodal problems. See [YVG20] for some examples you might consider. By your own account, the only real model/data examples you have (for logistic regression) do not show any benefit to using your method.
> > > >
> > > > Finally, "weights particles" was a typo: it should have been "weighted particles". I've corrected the error in my original comment.
> > > >
> > > > [SSK12] Sugiyama, M., Suzuki, T. & Kanamori, T. Density-ratio matching under the Bregman divergence: a unified framework of density-ratio estimation. Ann Inst Stat Math 64, 1009–1044 (2012).
> > > > [YVG20] Yao, Y., Vehtari, A. & Gelman, A. Stacking for Non-mixing Bayesian Computations: The Curse and Blessing of Multimodal Posteriors. arXiv.org, arXiv:2006.12335 (2020).

---

> > > > > ### Author Response · Authors · 2021-11-30
> > > > > **Response to the additional comments by Reviewer g4HT**
> > > > >
> > > > > "Thank you for your thorough response. However, you seem to be missing my central concerns and claims: In your response regarding the 2-Wasserstein gradient flow for the relative entropy, you never refute my central claim: whatever particular representation you may choose to use (in terms of $X_t$ or $q_t$, for example), its unique solution is a Langevin diffusion. How can you then not mention this crucial fact – that you are approximating the Langevin diffusion – in your paper? My reference to Eq. (4.1) in Pavliotis (2014) was just to the form of the Fokker--Planck equation for the Langevin diffusion; I'm well aware that book is not about gradient flows. My reference to Liu (2017) was just an ML reference to the fact that the solution to the 2-Wasserstein gradient flow for the relative entropy is the Langevin diffusion."
> > > > >
> > > > > Your so-called "central claim" is not substantively relevant to our method and you really misunderstood our work. In terms of the particles $X_t$, the ordinary differential equation (ODE) representation of the gradient flow we use is different from the Ito stochastic differential equation (SDE) representation of the Langevin diffusion, as we have clarified in our previous response. In terms of the density functions $q_t$, the 2-Wasserstein gradient flow for the relative entropy is known to satisfy a linear Fokker-Planck equation (i.e., the forward Kolmogorov equation for the Langevin diffusion), which is also a continuity equation with specific velocity fields. In terms of particles $X_t$, the gradient flow can be represented as an Ito SDE, which corresponds to the Langevin diffusion. However, in terms of particle $X_t$, the gradient flow can also be represented as an ODE with velocity fields specified in the continuity equation. In our work, we discretize the ODE with the forward Euler method, which leads to a deterministic approach to updating the particles $X_t$.
> > > > >
> > > > >
> > > > > "You are right that what you are doing is not quite reverse logistic regression – I should have double checked that. Rather, your approach falls under the framework described in [SSK12] (as does reverse logistic regression). In any case, my main point stands: high-dimensional density (ratio) estimation and normalization constant estimation are difficult problems. If your method requires solving either, then the methods you compare against should compete on a level playing field. Hence, you should compare to using MALA with stacking or something similar."
> > > > >
> > > > > We agree in general high-dimensional density ratio estimation is a difficult problem. However, although our problem has connection to the standard density ratio estimation problem, our setting is different and we did not attempt to estimate normalization constant. In our work, we use a nonparametric method to estimate the density ratio between an unnormalized density (known up to a normalizing constant) and an underlying density of the particles. In our method, we do not need to know the normalizing constant because the velocity fields only involve the gradients of the log density ratios . Please read our paper just a little more carefully before making these inaccurate comments.
> > > > >
> > > > > There are many existing sampling methods in the literature. We compare our method with some of the most important and more widely known and tested methods. Of course, we would be happy to compare with MALA with stacking as suggested, even though we are not sure if this is a widely used and well-tested method.
> > > > >
> > > > > "Regarding your method's performance for a multimodal distribution, I'm not claiming it didn't perform well on the simple, 2D synthetic examples from your paper. However, (A) the comparison to a naive MALA implementation (see my #4 above) is not appropriate and (B) this result says almost nothing about how the method would perform on realistic multimodal problems. See [YVG20] for some examples you might consider. By your own account, the only real model/data examples you have (for logistic regression) do not show any benefit to using your method."
> > > > >
> > > > > Thank you for suggesting this reference with some additional example. We will try our method on these examples.
> > > > >
> > > > > "Finally, "weights particles" was a typo: it should have been ``weighted particles". I've corrected the error in my original comment."
> > > > >
> > > > > Thanks for this clarification. However, we do not use the so-called "weighted particles".  As described in Algorithm 1 in our manuscript, particles are initialized as i.i.d. samples from the source distribution $q_0$ and get updated with the forward Euler iterations. No weights are placed on the particles.

---

### Decision · Program_Chairs · 2022-01-20

**Decision:**

Reject

**Comment:**

The paper proposes a sampling technique for unnormalized distributions. The main idea is to gradually transform particles by following the gradient flow of the relative entropy in the Wasserstein space of probability distributions. The paper tackles an important problem and provides an interesting new perspective. However, even putting aside the concerns on the theoretical analysis raised by the reviewers, the experimental evaluations does not seem sufficient to demonstrate the benefits of the proposed approach.